# A Formal Comparison Between Chain of Thought and Latent Thought

Kevin Xu [1]   Issei Sato [1]

## Abstract

Chain of thought (CoT) elicits reasoning in large language models by explicitly generating intermediate tokens. In contrast, latent thought reasoning operates directly in the continuous latent space, enabling computation beyond discrete linguistic representations. While both approaches exploit iterative computation, their comparative capabilities remain underexplored. In this work, we present a formal analysis showing that latent thought admits more efficient parallel computation than inherently sequential CoT. In contrast, CoT enables approximate counting and sampling through stochastic decoding. These separations suggest the tasks for which depth-driven recursion is more suitable, thereby offering practical guidance for choosing between reasoning paradigms. Code is available at https://github.com/kevin671/cot-vs-loop.

## 1. Introduction

Transformer-based large language models (LLMs) (Vaswani et al., 2017) have shown strong performance across diverse tasks and have recently been extended to complex reasoning. Rather than directly predicting final answers, generating intermediate reasoning steps, known as chain of thought (CoT) (Wei et al., 2022), enhances reasoning abilities. This naturally raises the question: *why is CoT effective for complex tasks?* Recent studies have approached this question by framing reasoning as a computational problem and analyzing its complexity (Feng et al., 2023; Merrill & Sabharwal, 2024; Li et al., 2024; Nowak et al., 2024), showing that CoT improves performance by increasing the model's effective depth through iterative computation, thereby enabling the solution of problems that would otherwise be infeasible.

As an alternative to CoT, recent work has explored latent

[1]Department of Computer Science, The University of Tokyo, Japan. Correspondence to: Kevin Xu <kevinxu@g.ecc.u-tokyo.ac.jp>, Issei Sato <sato@g.ecc.u-tokyo.ac.jp>.

*Proceedings of the 43rd International Conference on Machine Learning*, Seoul, South Korea. PMLR 306, 2026. Copyright 2026 by the author(s).

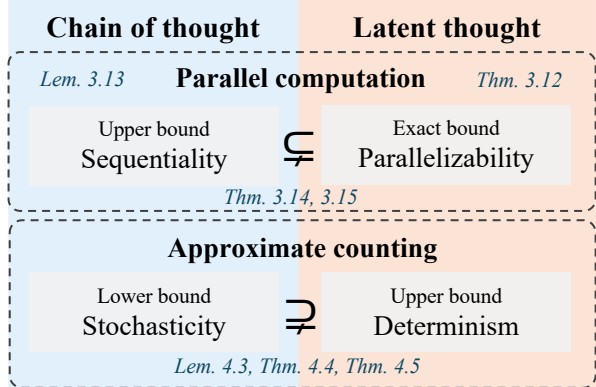

*Figure 1.* Overview of our formal comparison of the expressive power of chain-of-thought and latent thought reasoning.

thought, which reasons directly in the hidden state space rather than in the discrete token space. This paradigm includes chain of continuous thought (Coconut) (Hao et al., 2025), which replaces next tokens with hidden state, and *looped Transformer (looped TF)*, in which output hidden states are iteratively fed back as inputs (Dehghani et al., 2019). Such iterative architectures have been shown to enhance expressivity: Coconut enables the simultaneous exploration of multiple traces (Zhu et al., 2025a; Gozeten et al., 2025), while looped TF satisfies universality (Giannou et al., 2023; Xu & Sato, 2025) and demonstrates competitive empirical performance (Csordás et al., 2024; Bae et al., 2025; Zhu et al., 2025b).

These reasoning paradigms share the core idea of iteratively applying Transformers to enhance expressive power, which naturally leads to a fundamental question:

> *What is the separation between*
> *chain of thought and latent thought?*

Recent studies characterize how expressivity scales with the number of iterations. Specifically, it has been shown that looped TF subsumes deterministic CoT (Saunshi et al., 2025), and exhibits a strict separation with only a logarithmic number of iterations (Merrill & Sabharwal, 2025a). Nevertheless, fundamental questions remain open:

> *Does the separation extend beyond the logarithmic regime?*
> *Is latent thought always more expressive than CoT?*

## 1.1. Our Contributions

In this work, we address both questions by clarifying the respective strengths and limitations of the two reasoning paradigms through a formal complexity-theoretic analysis of their expressive power. Specifically, we show that latent thought gains efficiency from its parallelizability, yielding separations beyond the polylogarithmic regime. In contrast, CoT benefits from stochasticity, which enables approximate counting. An overview is given in Fig. 1.

**Latent thought enables parallel reasoning.** By formalizing decision problems as the evaluation of directed acyclic graphs (DAGs), we reveal the parallel computational capability of latent thought utilizing continuous hidden states. This analysis can be formalized by relating the class of decision problems realizable by the model to Boolean circuits. In particular, Boolean circuits composed of logic gates such as AND, OR, NOT, and Majority, with polylogarithmic depth $\log^k n$ for $k \in \mathbb{N}$ and input size $n$, define the class $\mathsf{TC}^k$, a canonical model of parallel computation. Circuit complexity plays a central role in analyzing the computational power of Transformer models: fixed-depth Transformers without CoT are known to be upper-bounded by $\mathsf{TC}^0$ (Merrill & Sabharwal, 2023), and subsequent studies analyze how the expressivity of CoT scales their computational power in terms of Boolean circuit complexity (Li et al., 2024). We show that latent thought with $\log^k n$ iterations exactly captures the power of $\mathsf{TC}^k$ (Thm. 3.12); in contrast, CoT with $\log^k n$ steps cannot realize the full power of $\mathsf{TC}^k$ (Thm. 3.13) due to its inherent sequentiality. This yields a strict separation in favor of latent thought in polylogarithmic regime (Thm. 3.15), showing its efficiency in terms of the required number of iterations.

**CoT enables approximate counting.** A counting problem is a fundamental task in mathematics and computer science that determines the number of solutions satisfying a given set of constraints, including satisfying assignments of Boolean formulas, graph colorings, and partition functions (Arora & Barak, 2009). While exact counting for the complexity class $\#\mathsf{P}$ is generally computationally intractable, approximation provides a feasible alternative. We show that CoT supports fully polynomial-time randomized approximation schemes (FPRAS), yielding reliable estimates even in cases where exact counting via deterministic latent thought reasoning is intractable (Lemma 4.3). Furthermore, leveraging classical results connecting approximate counting and sampling (Jerrum et al., 1986), we extend this separation to distribution modeling: there exist target distributions that CoT can approximately represent and sample from, but that remain inaccessible to latent thought (Theorem 4.4). To the best of our knowledge, this constitutes the first formal separation in favor of CoT.

## 2. Background

### 2.1. Models of Computation

We define a class of reasoning paradigms in which a Transformer block (Vaswani et al., 2017) is applied iteratively. Informally, CoT generates intermediate reasoning steps explicitly as tokens in an autoregressive manner. Formal definitions and illustrations are given in Appendix A.

**Definition 2.1** (CoT, following Merrill & Sabharwal (2024))**.** Let $\mathcal{V}$ be a vocabulary, and let $\mathrm{TF}_{\mathrm{dec}} : \mathcal{V}^* \to \mathcal{V}$ denote an decoder-only Transformer. Given an input sequence $x = (x_1, \ldots, x_n) \in \mathcal{V}^n$, the outputs of *CoT* are defined by

$$f_{\mathrm{cot}}^0(x) := x, \quad f_{\mathrm{cot}}^{k+1}(x) := f_{\mathrm{cot}}^k(x) \cdot \mathrm{TF}_{\mathrm{dec}}(f_{\mathrm{cot}}^k(x)),$$

where $\cdot$ denotes concatenation. We define the output to be the last tokens of $f_{\mathrm{cot}}^{T(n)}(x) \in \mathcal{V}^{n+T(n)}$.

Coconut feeds the final hidden state as the embedding of the next token. Although the original Coconut model (Hao et al., 2025) can generate both language tokens and hidden states, we focus exclusively on hidden state reasoning steps, in order to compare its representational power with that of CoT. Here, $\mathbb{F}$ denotes the set of finite-precision floating-point numbers, and $d \in \mathbb{N}$ denotes the embedding dimension.

**Definition 2.2** (Coconut)**.** Let $\mathcal{V}$ be a vocabulary and let $\mathrm{TF}_{\mathrm{dec}}^{\mathrm{Coconut}} : \mathcal{V}^* \times (\mathbb{F}^d)^* \to \mathbb{F}^d$ be a decoder-only Transformer that maps a fixed token prefix together with a hidden state to the next hidden state. Given an input sequence $x = (x_1, \ldots, x_n) \in \mathcal{V}^n$, we define the hidden states recursively by

$$h^0 := \left(e(x_i)\right)_{i=1}^n, \quad h^{k+1} := \mathrm{TF}_{\mathrm{dec}}^{\mathrm{Coconut}}(x, h^k),$$

where $e : \mathcal{V} \to \mathbb{F}^d$ denotes an embedding. The output after $T(n)$ steps is obtained by decoding a suffix of the hidden state sequence ending at $h^{T(n)}$.

Looped TFs, by contrast, feed the entire model output back into the input without generating explicit tokens, recomputing all hidden states of the sequence at every iteration.

**Definition 2.3** (Looped TF)**.** Let $\mathrm{TF} : \mathbb{F}^{d \times *} \to \mathbb{F}^{d \times *}$ denote a Transformer block. Given an input sequence $x = (x_1, \ldots, x_n) \in \mathcal{V}^n$, the outputs are defined recursively by

$$f_{\mathrm{loop}}^0(x) := \left(e(x_i)\right)_{i=1}^n, \quad f_{\mathrm{loop}}^{k+1}(x) := \mathrm{TF}(f_{\mathrm{loop}}^k(x)),$$

where $e : \mathcal{V} \to \mathbb{F}^d$ denotes an embedding. The output after $T(n)$ loop iterations is the decoded last tokens of $f_{\mathrm{loop}}^{T(n)}(x)$.

Here, we assume that the input for looped TF may include sufficient padding so that its length is always at least as large as the output length, as in (Merrill & Sabharwal, 2025b). The definitions of the models describe their core architectures; the specific details may vary depending on the tasks to which they are applied.

*Table 1.* Comparison between prior theoretical analyses and our work on the computational power of CoT, Coconut, and looped TF.

| Paper | Model | Det | Pro | Class | Problem Setting |
|---|---|---|---|---|---|
| Li et al. (2024) | CoT | ✓ | - | ✓ | Boolean circuit |
| Nowak et al. (2024) | CoT | - | ✓ | ✓ | Language modeling |
| Saunshi et al. (2025) | CoT / Loop | ✓ | - | - | Group composition |
| Merrill & Sabharwal (2025b) | CoT / Loop | ✓ | - | ✓ | Uniform $TC^k$ |
| Gozeten et al. (2025); Zhu et al. (2025a) | CoT / Coconut | ✓ | - | - | Graph exploration |
| Svete & Sabharwal (2025) | CoT / Loop / Masked | ✓ | - | ✓ | Uniform $TC^k$ |
| **Ours** | CoT / Coconut / Loop | ✓ | ✓ | ✓ | $TC^k$ & Approximate counting |

## 2.2. Related Work

To understand the expressive power of Transformers, previous work studies which classes of problems can be solved and with what computational efficiency. These questions can be naturally analyzed within the framework of computational complexity theory. Such studies on CoT and latent thought are summarized below and in Table 1.

**Computational power of chain of thought.** The expressivity of Transformers is limited by bounded depth (Merrill & Sabharwal, 2023), whereas CoT enhances their expressiveness by effectively increasing the number of sequential computational steps, enabling the solution of problems that would otherwise be intractable for fixed-depth architectures (Feng et al., 2023). Recent work has investigated how the expressivity of CoT scales with the number of reasoning steps, formalizing CoT for decision problems and computational complexity classes (Merrill & Sabharwal, 2024; Li et al., 2024). Beyond decision problems, CoT has been further formalized in a probabilistic setting for representing probability distributions over strings (Nowak et al., 2024).

**Computational power of latent thought.** Latent thought is an alternative paradigm for increasing the number of computational steps without being constrained to the language space, with the potential to enhance model expressivity. In particular, Coconut has been shown to enable the simultaneous exploration of multiple candidate reasoning traces (Zhu et al., 2025a; Gozeten et al., 2025). Looped TFs can simulate iterative algorithms (Yang et al., 2024; de Luca & Fountoulakis, 2024) and, more generally, realize polynomial-time computations (Giannou et al., 2023). Recent results further demonstrate advantages over chain of thought reasoning: looped TFs can subsume the class of deterministic computations realizable by CoT using the same number of iterations (Saunshi et al., 2025), and exhibit a strict separation within the same logarithmic iterations (Merrill & Sabharwal, 2025a). Concurrent work (Zhu et al., 2025a; Merrill & Sabharwal, 2025b) has also identified connections between parallel reasoning and Coconut and diffusion models.

## 3. Latent Thought Enables Parallel Reasoning

We formalize the reasoning problem as a graph evaluation problem. Section 3.2 illustrates how each model approaches the same problem differently, providing intuitive insight into their contrasting capabilities. Building on these observations, Section 3.3 characterizes their expressive power and establishes a formal separation between them.

### 3.1. Problem Setting

Reasoning problems that can be solved by straight-line programs admit representations as directed acyclic graphs (DAGs) (Aho & Ullman, 1972), as illustrated in Fig. 2(a).

**Definition 3.1** (Computation graph). Let $\Sigma$ be a finite alphabet, and let $\mathcal{F}$ denote a finite set of functions $f : \Sigma^* \to \Sigma$. A *computation graph* is a directed acyclic graph $G_n = (V_n, E_n)$ that defines a function $F_{G_n} : \Sigma^n \to \Sigma^{m(n)}$, where $m(n)$ denotes the output length. Here $V_n$ denotes the set of nodes, consisting of (i) $n$ input nodes with in-degree $0$, (ii) function nodes labeled by $f \in \mathcal{F}$, which take as arguments the predecessor nodes specified by their incoming edges in $E_n$, and (iii) $m(n)$ output nodes with out-degree $0$. The overall function is obtained by evaluating the graph in topological order. The *size* of the graph is $|V_n|$, denoted by $\text{size}(G_n)$, and its *depth* is the length of the longest path from an input to an output node, denoted by $\text{depth}(G_n)$.

**Assumptions on models.** Our goal is to evaluate the computational efficiency of each model via an asymptotic analysis of how the required number of reasoning steps or loops scales with the input size $n$. Beyond *time complexity*, we also allow the *space complexity* of the model to scale with the input size $n$. In particular, the embedding dimension in Transformer blocks can be viewed as analogous to the number of processors in classical parallel computation models. Accordingly, we adopt a *non-uniform* computational model, in which a different model is allowed for each input size. This non-uniform setting is standard in the study of circuit complexity and parallel computation (Cook, 1985), and is consistent with prior analyses of Transformers and CoT (Sanford et al., 2024b; Li et al., 2024).

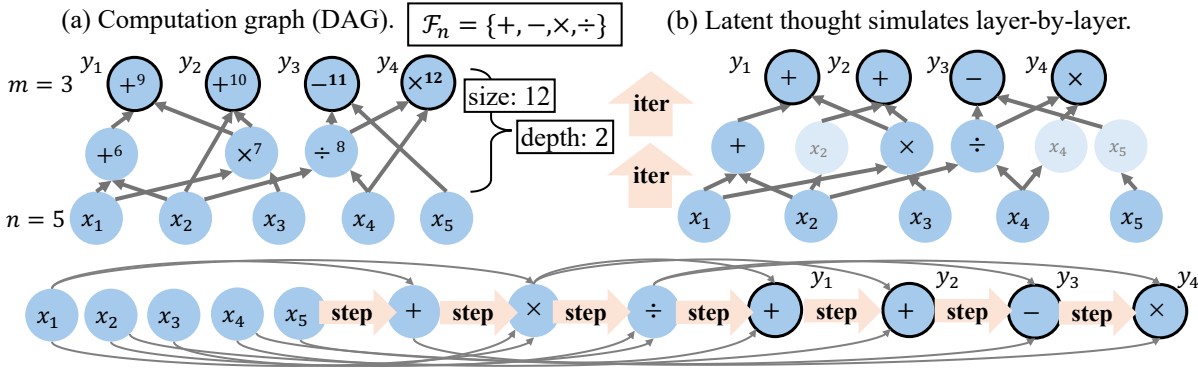

*Figure 2.* Comparison of reasoning paradigms for evaluating a DAG. **(a)** A computation graph $G_n$. **(b)** Latent thought can simulate the computation layer by layer in parallel, using a number of loops equal to the depth of the graph, $\text{depth}(G_n)$. **(c)** CoT can sequentially simulate the computation node by node, using a number of steps proportional to the size of the graph, $O(\text{size}(G_n))$.

**On the fairness of comparing steps and loops.** We analyze expressivity in terms of the number of reasoning steps. Although this may appear unfair in terms of raw computation, it is justified when comparing latency. Specifically, CoT benefits from KV caching, which makes each step computationally inexpensive; however, accessing cached states is typically memory-bound, leaving compute resources underutilized. In contrast, looped TFs recompute over the full sequence at each iteration, incurring higher arithmetic cost but achieving higher arithmetic intensity and better utilization of modern parallel hardware. As a result, the latency of looped TFs is comparable to that of CoT.

### 3.2. CoT Suffices with Size-scaled Steps and Latent Thought Suffices with Depth-scaled Iterations

We show how each model can evaluate DAGs, which provides a lower bound on their expressivity and offers intuition for the distinctions between the models, in terms of sequentiality and parallelizability. Before presenting our main result, we first state the underlying assumptions.

**Definition 3.2** (Merrill & Sabharwal, 2023)**.** The model is *log-precision*, where each scalar is stored with $O(\log n)$ bits and every arithmetic operation is rounded to that precision.

**Assumption 3.3** (Poly-size graph)**.** $\text{size}(G_n) \in \text{poly}(n)$.

**Assumption 3.4** (Poly-efficient approximation, cf. (Feng et al., 2023))**.** Each node function of $G_n$ can be approximated by a log-precision feedforward network whose parameter size is polynomial in the input length and the inverse of the approximation error. We denote by $\text{ff\_param}(G_n)$ an upper bound such that every $f \in \mathcal{F}$ admits such a network with at most $\text{ff\_param}(G_n)$ parameters.

Under these assumptions, we show that CoT can simulate computation by sequentially decoding nodes, where intermediate tokens serve as a scratchpad allowing the evaluation of each node once all its predecessors have been computed.

**Theorem 3.5** (CoT for DAGs)**.** *Let* $\{G_n\}_{n \in \mathbb{N}}$ *be a family of computation graphs that satisfy Assumptions 3.3 and 3.4. Then, for each* $n \in \mathbb{N}$*, there exists a log-precision CoT with parameter size bounded by* $O(\text{ff\_param}(G_n))$*, such that for every input* $x \in \Sigma^n$*, the model outputs* $F_{G_n}(x)$ *with steps proportional to the "size" of the graph, i.e.,* $O(\text{size}(G_n))$*.*

*Proof sketch.* At each step, the attention layer retrieves the outputs of predecessor nodes from previously generated tokens, and a feed-forward layer then computes the node function, whose output is generated as the next token. □

In contrast, latent thought can operate in parallel, layer by layer, where all nodes at the same depth are computed simultaneously, provided that the model has sufficient size.

**Theorem 3.6** (Latent thought for DAGs)**.** *Let* $\{G_n\}_{n \in \mathbb{N}}$ *be a family of computation graphs that satisfy Assumptions 3.3 and 3.4. Then, for each* $n \in \mathbb{N}$*, there exists a log-precision Coconut and looped TF with parameter size* $O(\text{ff\_param}(G_n) \cdot \text{size}(G_n))$*, such that for every input* $x \in \Sigma^n$*, it computes* $F_{G_n}(x)$ *with iterations proportional to the "depth" of the graph* $G_n$*, i.e.,* $O(\text{depth}(G_n))$*.*

*Proof sketch.* The role assignment of each layer is based on (Li et al., 2024), as shown in Figure 9. An attention layer aggregates its inputs into a single hidden state. Unlike discrete tokens, continuous latent states allow the simultaneous encoding of the outputs of multiple nodes, enabling the feed-forward layer to compute node functions in parallel. □

**Remark.** Illustrations are provided in Fig. 2, with formal proofs deferred to Appendix B. These results reveal distinct characteristics: CoT can utilize intermediate steps as scratchpad memory and perform computations sequentially, whereas latent thought can leverage structural parallelism to achieve greater efficiency with sufficient resources.

## 3.3. Separation in Polylogarithmic Iterations

In this section, we shift to formal decision problems to precisely characterize the computational power of each reasoning paradigm, clarify what cannot be achieved, and use these limitations to derive rigorous separations. We begin by defining their complexity classes, as in (Li et al., 2024).

**Definition 3.7** (Complexity Classes CoT, CT and Loop). Let $\mathsf{CoT}[T(n), d(n), s(n)]$, $\mathsf{CT}[T(n), d(n), s(n)]$, and $\mathsf{Loop}[T(n), d(n), s(n)]$ denote the sets of languages $\mathcal{L} : \{0,1\}^* \to \{0,1\}$ for which there exists a *deterministic* CoT, Coconut, and looped TF, respectively, denoted by $M_n$ for each input size $n$, with embedding size $O(d(n))$ and $O(s(n))$ bits of precision, such that for all $x \in \{0,1\}^n$, the final output token after $O(T(n))$ iterations equals $\mathcal{L}(x)$.

Boolean circuits serve as a standard formal model of computation, where processes are defined by the evaluation of DAGs with well-established complexity measures.

**Definition 3.8** (Informal). A *Boolean circuit* is a DAG over the alphabet $\Sigma = \{0,1\}$, where each internal node (gate) computes a Boolean function such as AND, OR, or NOT. $\mathsf{SIZE}[s(n)]$ and $\mathsf{DEPTH}[d(n)]$ denote the class of languages decidable by a non-uniform circuit family $\{C_n\}$ with size $O(s(n))$ and depth $O(d(n))$, respectively.

First, we formalize the results of the previous section to show that latent thought iterations can represent circuit depth, whereas CoT corresponds to circuit size.

**Theorem 3.9** (Li et al., 2024). $\forall T(n) \in \mathrm{poly}(n)$,

$$\mathsf{SIZE}[T(n)] \subseteq \mathsf{CoT}[T(n), \log n, 1].$$

**Theorem 3.10.** *For any function $T(n) \in \mathrm{poly}(n)$ and any non-uniform circuit family $\{C_n\}$, it holds that*

$$\mathsf{DEPTH}[T(n)] \subseteq \mathsf{Loop}[T(n), \mathrm{size}(C_n), 1],$$
$$\mathsf{DEPTH}[T(n)] \subseteq \mathsf{CT}[T(n), \mathrm{size}(C_n), 1].$$

Boolean circuits serve as a formal model of parallel computations that run in polylogarithmic time using a polynomial number of processors (Stockmeyer & Vishkin, 1984).

**Definition 3.11.** For each $k \in \mathbb{N}$, the classes $\mathsf{NC}^k$, $\mathsf{AC}^k$, and $\mathsf{TC}^k$ consist of languages decidable by non-uniform circuit families of size $\mathrm{poly}(n)$ and depth $O(\log^k n)$, using bounded-fanin Boolean gates, unbounded-fanin AND/OR gates, and threshold gates, respectively.

We then characterize the exact computational power of latent thought in the parallel computation regime.

**Theorem 3.12.** *For each $k \in \mathbb{N}$, it holds that*

$$\mathsf{Loop}[\log^k n, \ \mathrm{poly}(n), \ 1 \ (\text{resp. } \log n)]$$
$$= \mathsf{CT}[\log^k n, \ \mathrm{poly}(n), \ 1 \ (\text{resp. } \log n)]$$
$$= \mathsf{AC}^k \ (\text{resp. } \mathsf{TC}^k).$$

Figure 3. The separation between latent thought and CoT for decision problems, under polylogarithmic iterations.

*Proof sketch.* The inclusion from circuits to latent thought follows from Theorem 3.10. For the converse inclusion, we build on the arguments of prior work (Merrill & Sabharwal, 2023; Li et al., 2024), which show that a fixed-depth Transformer block under finite precision is contained in $\mathsf{AC}^0$ (or $\mathsf{TC}^0$ under logarithmic precision). We extend their analysis to the looped setting, which can be unrolled into a $\mathsf{TC}^k$ circuit by composing a $\mathsf{TC}^0$ block for $\log^k n$ iterations. □

Moreover, we establish an upper bound on the power of CoT in the parallel computation regime. This limitation arises from the inherently sequential nature of CoT.

**Lemma 3.13.** *For each $k \in \mathbb{N}$, it holds that*

$$\mathsf{CoT}[\log^k n, \mathrm{poly}(n), \log n] \subseteq \mathsf{TC}^{k-1}.$$

*Proof.* The total $\log^k n$ steps can be divided into $\log^{k-1} n$ blocks, each consisting of $\log n$ steps. Since $\mathsf{CoT}[\log n, \mathrm{poly}(n), \log n] \subseteq \mathsf{TC}^0$ (Li et al., 2024), each block with the previous block's outputs fed as inputs to the next block can be simulated in $\mathsf{TC}^0$; iterating this over $\log^{k-1} n$ layers yields a circuit in $\mathsf{TC}^{k-1}$. □

These results lead to a separation in expressive power under standard complexity assumptions, as illustrated in Figure 3.

**Theorem 3.14.** *For each $k \in \mathbb{N}$, if $\mathsf{TC}^{k-1} \subsetneq \mathsf{NC}^k$, then*

$$\mathsf{CoT}[\log^k n, \mathrm{poly}(n), \log n] \subsetneq \mathsf{Loop}[\log^k n, \mathrm{poly}(n), 1],$$
$$\mathsf{CoT}[\log^k n, \mathrm{poly}(n), \log n] \subsetneq \mathsf{CT}[\log^k n, \mathrm{poly}(n), 1].$$

**Theorem 3.15.** *For each $k \in \mathbb{N}$, if $\mathsf{TC}^{k-1} \subsetneq \mathsf{TC}^k$, then*

$$\mathsf{CoT}[\log^k n, \mathrm{poly}(n), \log n] \subsetneq \mathsf{Loop}[\log^k n, \mathrm{poly}(n), \log n],$$
$$\mathsf{CoT}[\log^k n, \mathrm{poly}(n), \log n] \subsetneq \mathsf{CT}[\log^k n, \mathrm{poly}(n), \log n].$$

**Remark.** The claims follow directly from Theorem 3.12 and Lemma 3.13. The established separations of the complexity classes, as summarized in Fig. 3, show that latent thought reasoning enables efficient parallel solutions more effectively than CoT, which is inherently sequential.

# 4. CoT Enables Approximate Counting

In the previous section, we showed that for decision problems, latent thought can yield more efficient solutions than CoT. This naturally raises the question of whether latent thought is universally more powerful than CoT. While prior work has shown that CoT can be simulated by looped Transformer models for deterministic decision problems under deterministic decoding (Saunshi et al., 2025), we found that this result does not directly extend to probabilistic settings with stochastic decoding. Accordingly, we shift our focus from efficiency in terms of the number of reasoning steps to expressive capability under polynomially many iterations.

## 4.1. Preliminaries

**Approximate counting.** Formally, let $\Sigma$ be a finite alphabet and let $R \subseteq \Sigma^* \times \Sigma^*$ be a relation. For an input $x \in \Sigma^*$, define $R(x) := \{\, y \in \Sigma^* \mid (x, y) \in R \,\}$, and the counting problem is to determine $|R(x)|$. A wide class of natural relations admits a recursive structure, which allows solutions to be constructed from smaller subproblems.

**Definition 4.1** (Informal: Self-reducibility (Schnorr, 1976)). A relation $R$ is *self-reducible* if there exists a polynomial-time procedure that, given any input $x$ and prefix $y_{1:k}$ (with respect to a fixed output order), produces a sub-instance $\psi(x, y_{1:k})$ such that every solution $z$ of $\psi(x, y_{1:k})$ extends $y_{1:k}$ to a solution of $R(x)$ (and conversely), i.e., $R\big(\psi(x, y_{1:k})\big) = \{z \mid \operatorname{concat}(y_{1:k}, z) \in R(x)\}$.

While exact counting is intractable, there exist efficient randomized approximation algorithms (Karp & Luby, 1983).

**Definition 4.2** (FPRAS). An algorithm is called a *fully polynomial-time randomized approximation scheme* (FPRAS) for a function $f$ if, for any $\varepsilon > 0$ and $\delta > 0$, it outputs an estimate $\hat{f}(x)$ such that

$$\Pr\Big[(1 - \varepsilon)f(x) \le \hat{f}(x) \le (1 + \varepsilon)f(x)\Big] \ge 1 - \delta,$$

and runs in time polynomial in $|x|$, $1/\varepsilon$, and $\log(1/\delta)$.

The class of counting problems that admit an FPRAS is denoted by FPRAS. Although randomized algorithms provide only probabilistic guarantees, they are often both more efficient and simpler than their deterministic counterparts, denoted by FPTAS (Definition C.11). For example, counting the number of satisfying assignments of a DNF formula admits an FPRAS based on Monte Carlo methods (Karp et al., 1989), whereas no FPTAS is known for this problem. Moreover, probabilistic analysis enables us to capture algorithmic behavior on typical instances arising in real-world applications (Mitzenmacher & Upfal, 2017).

**Probabilistic models of computation.** In contrast to the deterministic models considered in the previous section,

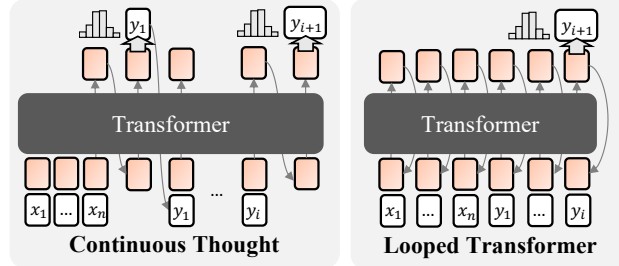

*Figure 4.* Probabilistic models of computation with latent thought. Each output token $y_i$ is stochastically generated.

we now study probabilistic models that define a conditional distribution over output strings $y = (y_1, \ldots, y_m) \in \Sigma^*$. We consider autoregressive next-token prediction of the form

$$p(y \mid x) = \prod_{i=1}^{m} p(y_i \mid x, y_{<i}),$$

where the model is allowed to perform additional reasoning steps before producing each output token $y_i$. This formulation was first used to formalize CoT for language modeling by Nowak et al. (2024). We also allow stochastic decoding for latent reasoning at the token level: reasoning iterations are performed entirely in hidden space and no linguistic tokens are sampled except for the output token $y_i$, as illustrated in Fig. 4. This definition is consistent with practical implementations (Csordás et al., 2024; Bae et al., 2025). Within this framework, we define complexity classes of probabilistic models, denoted by pCoT, pCT, and pLoop, respectively. Formal definitions are in Appendix C.1.

## 4.2. Separation in Approximate Counting

We first analyze the expressivity of the token-level conditional prediction at each step, $p(y_i \mid x, y_{<i})$, and show that CoT is strictly more expressive than latent thought in this setting. The key distinction is whether intermediate computation permits sampling. CoT explicitly samples intermediate reasoning tokens, inducing stochastic computation and enabling the emulation of randomized algorithms. In contrast, latent thought performs only deterministic transformations in latent space, resulting in deterministic computation.

**Lemma 4.3** (Informal). *Assume that* $\mathsf{FPTAS} \subsetneq \mathsf{FPRAS}$ *for self-reducible relations. There exists a self-reducible relation $R$ and an associated function $f : \Sigma^* \times \Sigma^* \to \mathbb{N}$ defined by $f(x, y_{<i}) := |\{\, z \in \Sigma^* : (x, y_{<i}z) \in R \,\}|$ such that CoT with polynomially many steps admits an FPRAS for $f$. Whereas, no latent thought with polynomially many iterations admits the same approximation guarantee.*

*Proof sketch.* For self-reducible relations, approximating the counting function $f$ on subproblems is polynomial-time inter-reducible with approximating $|R(x)|$ (Jerrum et al.,

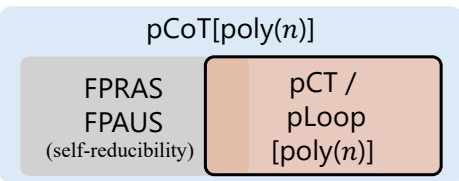

*Figure 5.* The separation for approximate counting (sampling).

1986). If latent thought with polynomially many iterations admitted an FPTAS for $f$, then it would induce a deterministic FPTAS for $|R(x)|$, contradicting the assumption. □

### 4.3. Separation in Approximate Sampling

We then move from token-level conditional distributions $p(y_i \mid x, y_{<i})$ to the full sequence-level distribution $p(y \mid x)$. Beyond approximate counting, we establish a separation for approximate sampling problems. Specifically, we construct target distributions for which the complexity of each conditional can be reduced to approximate counting.

**Theorem 4.4.** *Assume that* FPTAS $\subsetneq$ FPRAS *for self-reducible relations. There exists a distribution $p(y \mid x)$ over $y \in \Sigma^*$ and $x \in \Sigma^n$ such that a CoT with a polynomial number of steps, whose induced output conditionals are denoted by $q(y_i \mid x, y_{<i})$, admits an FPRAS for approximating the conditional probabilities $p(y_i \mid x, y_{<i})$ for all $x \in \Sigma^n$, indices $i \geq 1$, and prefixes $y_{<i} := (y_1, \ldots, y_{i-1})$. In contrast, no latent thought with polynomially many iterations admits the same approximation guarantee.*

*Proof sketch.* Define the target distribution $p$ to be the uniform distribution supported on the solution set $R(x)$. We rely on the classical result that approximate sampling from the uniform distribution over solutions, captured by the class FPAUS, is polynomial-time inter-reducible with approximate counting for self-reducible relations (Jerrum et al., 1986). Let $U(\cdot \mid x)$ denote the uniform distribution over solutions of a self-reducible relation $R(x)$. This distribution admits an autoregressive factorization $U(y \mid x) = \prod_{i=1}^{m} p(y_i \mid x, y_{<i})$, where each conditional probability is given by $p(y_i \mid x, y_{<i}) = \frac{\left| \{ z \in \Sigma^* : (x, y_{1:i+1}z) \in R \} \right|}{\left| \{ z \in \Sigma^* : (x, y_{1:i}z) \in R \} \right|}$. We show that each conditional probability, expressed as a ratio of subproblem counts, reduces to approximate counting. Then, applying Lemma 4.3 to these conditionals yields the desired separation for approximate sampling. □

Consequently, we obtain the following separations in favor of CoT, as also shown in Fig. 5.

**Theorem 4.5.** *Assuming* FPTAS $\subsetneq$ FPRAS *for self-reducible relations, it holds that*

$$\forall \mathcal{M} \in \{\text{pCT}, \text{pLoop}\}, \quad \mathcal{M}[\text{poly}(n)] \subsetneq \text{pCoT}[\text{poly}(n)].$$

## 5. Experiments

In this section, we provide empirical validation of our theoretical results on tasks with well-characterized complexity. Specifically, we study parallelizable tasks to empirically validate the efficiency of latent thought predicted in Section 3, and approximate counting and sampling tasks to demonstrate the effectiveness of CoT as shown in Section 4.

### 5.1. Experimental Setting

**Fundamental algorithmic reasoning tasks.** We use four problems. (1) Word problems for finite non-solvable groups: given a sequence of generators, the task is to evaluate their composition, which is $\text{NC}^1$-complete (Barrington, 1986), also studied for Looped TF (Merrill & Sabharwal, 2025a). (2) $s$–$t$ connectivity (STCON): given a directed graph $G = (V, E)$ and two vertices $s, t \in V$, the task is to decide whether $t$ is reachable from $s$, which belongs to $\text{TC}^1$ (Gibbons & Rytter, 1989). (3) Arithmetic expression evaluation: given a formula consisting of $+, \times, -, /$ operations on integers, the task is to evaluate it. This problem is $\text{TC}^0$-reducible to Boolean formula evaluation (Feng et al., 2023), which is $\text{NC}^1$-complete (Buss, 1987). (4) Edit distance: given two strings $x$ and $y$, the task is to compute the minimum cost to transform $x$ into $y$. By reducing the dynamic programming formulation to shortest paths, this problem is in $\text{TC}^1$ (Apostolico et al., 1990).

**Approximate counting tasks.** We consider DNF counting and uniform sampling of graph colorings, both of which admit fully polynomial randomized approximation schemes for counting and sampling (FPRAS and FPAUS). Specifically, DNF counting admits an FPRAS via Monte Carlo sampling (Karp et al., 1989), while approximate counting and sampling of graph colorings admit an FPAUS based on rapidly mixing Markov chain Monte Carlo under suitable degree and color constraints (Jerrum, 1995).

**Training strategy.** Since our primary objective is to study expressive power, we allow flexibility in optimization and training strategies. For CoT models, training is performed with supervision from explicit sequential algorithms. For fewer CoT steps, we compare two strategies: uniformly selecting steps from the indices of the complete trajectory (Bavandpour et al., 2025), and stepwise internalization (distillation) methods (Deng et al., 2024). For latent thought, we observe that looped TF is easier to train than Coconut, and therefore adopt looped TF as our instantiation of latent thought, with curriculum learning applied to certain tasks.

### 5.2. Results

Table 2 reports results on parallelizable tasks, comparing latent thought and CoT under varying numbers of iterations.

*Table 2.* Accuracy (%) of CoT and looped TF on parallelizable tasks across different numbers of iterations. Here, $n$ denotes the problem size. For CoT, we report the best accuracy achieved across the two training strategies.

| Task | $n$ | Looped Transformer | | | | Chain of Thought | | | |
|---|---|---|---|---|---|---|---|---|---|
| | | 2 | 4 | 6 | 8 | 8 | 16 | 32 | 64 |
| Word Problem | 64 | 0.8 | 0.8 | **100.0** | **100.0** | 0.8 | 0.8 | **100.0** | **100.0** |
| Graph Connectivity | 32 | 80.8 | 95.8 | **99.0** | **99.0** | 81.0 | 81.4 | 88.2 | **100.0** |
| Arithmetic Evaluation | 32/16 | 43.7 | 99.4 | 99.5 | **99.7** | 47.3 | 47.6 | 48.2 | **82.5** |
| Edit Distance | 32/16 | 57.3 | 72.9 | 86.2 | **90.7** | 76.5 | 80.9 | 87.5 | **94.8** |

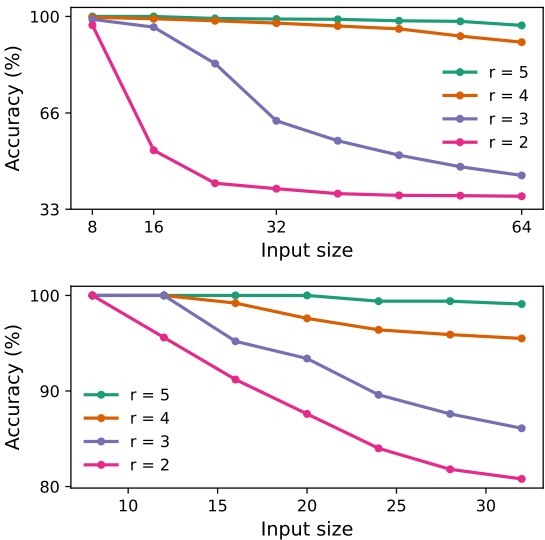

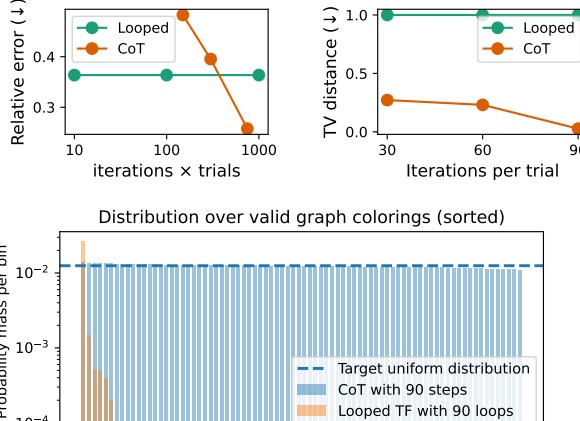

*Figure 7.* Top: Relative error for counting (left) and TV distance to uniform over valid colorings for sampling (right). Bottom: Empirical distributions for approximate sampling of graph colorings.

*Figure 6.* Accuracy of looped TFs on the arithmetic evaluation (top) and the connectivity (bottom). Each curve shows the performance for a fixed loop count $r$ as the input size $n$ increases.

Latent thought solves the problems with fewer iterations than CoT requires to reach comparable performance. These empirical results are consistent with our theoretical analysis: latent thought supports efficient parallel reasoning, in contrast to the inherently sequential nature of CoT. We also evaluate the relationship between performance, input size, and the number of iterations, as in prior studies (Sanford et al., 2024b; Merrill & Sabharwal, 2025a). Figure 6 presents our results for looped TFs, illustrating that as the input size $n$ increases, the number of loops required to maintain high accuracy grows only logarithmically, supporting our theoretical claim in the (poly-)logarithmic regime.

Figure 7 shows the results on the approximate counting or sampling tasks. For approximate counting, CoT performs Monte Carlo estimation: the effective number of samples is given by the product of the number of reasoning steps per trial and the number of independent trials. The probability mass plot illustrates how the empirical distribution over valid colorings compares to the target uniform distribution. We observe that CoT produces a distribution that is closer to

uniform, whereas the looped model concentrates probability mass on a smaller subset of solutions. This indicates that CoT achieves more uniform coverage of the valid colorings.

## 6. Conclusion

We formally analyze the computational capabilities of chain-of-thought and latent thought reasoning, providing a rigorous comparison that reveals their respective strengths and limitations. Specifically, we show that latent thought enables efficient parallel computation, whereas CoT enables randomized approximate counting. Our results provide practical guidance for selecting between reasoning paradigms: latent reasoning is more suitable for problems that can be solved efficiently, whereas CoT is more effective for more complex problems. For future work, an important direction is to investigate whether techniques such as distillation can reduce the number of iterations without compromising computational power. Another promising avenue is to extend our analysis to diffusion language models, which possess both parallelizability and stochasticity. Moreover, extending the analysis to realistic downstream tasks remains an important direction.

## Acknowledgements

This work was supported by JSPS KAKENHI Grant Number JP24H00709.

## Impact Statement

This paper presents work whose goal is to advance the field of Machine Learning. There are many potential societal consequences of our work, none which we feel must be specifically highlighted here.

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

# A. Formal Definitions

## A.1. Notation

Vectors are written in lowercase bold letters (e.g., $\boldsymbol{x}$) and matrices in uppercase bold letters (e.g., $\boldsymbol{W}$). The $i$-th entry of a vector $\boldsymbol{x}$ is $\boldsymbol{x}_i$, the vector from the $i$-th to the $j$-th entry is denoted by $\boldsymbol{x}_{i:j}$, and the $(i, j)$-th entry of a matrix $\boldsymbol{W}$ is $\boldsymbol{W}_{i,j}$. We use the symbol $*$ to denote a "don't care" value (or block of values). For $n \in \mathbb{N}^+$, let $[n] := \{1, 2, \ldots, n\}$. We sometimes write column vectors horizontally, e.g., $\boldsymbol{x} = (x_1, \ldots, x_n)$, for brevity. The Hadamard (element-wise) product is $\odot$. $\boldsymbol{e}_i \in \{0, 1\}^d$ is the $i$-th standard basis vector, $\boldsymbol{1}_d \in \mathbb{R}^d$ (or $1_d$) the all-ones vector, and $\boldsymbol{0}_d \in \mathbb{R}^d$ the zero vector. $\boldsymbol{I}_d \in \mathbb{R}^{d \times d}$ denotes the $d \times d$ identity matrix, and $\boldsymbol{0}_{m \times n} \in \mathbb{R}^{m \times n}$ the $m \times n$ zero matrix. The indicator function is $\boldsymbol{1}[\cdot]$, and $\bigoplus$ denotes block-diagonal concatenation. Functions on scalars or vectors are written in upright letters (e.g., FFN), while functions on matrices are boldface (e.g., $\mathbf{ATTN}$). Boldface is also used when scalar- or vector-level functions are extended to sequence level and applied independently to each token (e.g., $\mathbf{FFN}$). Finally, $\mathrm{poly}(n)$ denotes the set of functions growing at most polynomially: $\mathrm{poly}(n) := \left\{ f : \mathbb{N} \to \mathbb{N} \mid \exists k \in \mathbb{N}, \ \exists c > 0, \ \forall n \in \mathbb{N}, \ f(n) \leq c \cdot n^k \right\}$.

## A.2. Transformer Block

We define the computational components of a Transformer block using the notation of (Merrill & Sabharwal, 2025a). Let $\mathbb{F}_s$ denote the set of $s$-bit floating-point numbers with truncated arithmetic (Definition B.1).

**Definition A.1** (Transformer). A Transformer consists of the following components:

1. A word embedding function $\mathrm{WE} : \mathcal{V} \to \mathbb{F}_s^m$, where $\mathcal{V}$ denotes the vocabulary set.
2. A positional embedding function $\mathrm{PE} : \mathbb{N} \to \mathbb{F}_s^m$.
3. A multi-head self-attention layer $\mathbf{SA} : \mathbb{F}_s^{m \times N} \to \mathbb{F}_s^{m \times N}$ for arbitrary sequence length $N$, parameterized by a matrix $\boldsymbol{O} : \mathbb{F}_s^{s \times H} \to \mathbb{F}_s^m$ and, for each head $h \in [H]$ with head size $s$, matrices $\boldsymbol{Q}_h, \boldsymbol{K}_h, \boldsymbol{V}_h : \mathbb{F}_s^m \to \mathbb{F}_s^s$. Given an input $\boldsymbol{x}_i \in \mathbb{F}_s^m$ for each position $i \in [N]$, it computes the query $\mathbf{q}_{i,h} = \boldsymbol{Q}_h \boldsymbol{x}_i$, key $\boldsymbol{k}_{i,h} = \boldsymbol{K}_h \boldsymbol{x}_i$, and value $\mathbf{v}_{i,h} = \boldsymbol{V}_h \boldsymbol{x}_i$, and outputs $\boldsymbol{O} \cdot (\boldsymbol{a}_{i,1}, \ldots, \boldsymbol{a}_{i,H})$, where each attention output $\boldsymbol{a}_{i,h}$ is defined, for softmax function, as:

$$\boldsymbol{a}_{i,h} = \sum_{j=1}^{c(i)} \frac{\exp(\mathbf{q}_{i,h}^\top \boldsymbol{k}_{j,h})}{Z_{i,h}} \cdot \mathbf{v}_{j,h}, \quad Z_{i,h} = \sum_{j=1}^{c(i)} \exp(\mathbf{q}_{i,h}^\top \boldsymbol{k}_{j,h}), \tag{1}$$

   with $c(i) = i$ for causal attention and $c(i) = N$ for full attention. For the saturated hardmax attention (Merrill et al., 2022), each attention output $\boldsymbol{a}_{i,h}$ is defined as:

$$\boldsymbol{a}_{i,h} = \sum_{j \in M_{i,h}} \frac{1}{|M_{i,h}|} \mathbf{v}_{j,h}, \quad M_{i,h} = \left\{ j \in [c(i)] \ \middle| \ \mathbf{q}_{i,h}^\top \boldsymbol{k}_{j,h} = \max_{j'} \mathbf{q}_{i,h}^\top \boldsymbol{k}_{j',h} \right\}. \tag{2}$$

4. A feedforward layer $\mathrm{FF} : \mathbb{F}_s^m \to \mathbb{F}_s^m$ with parameter $\boldsymbol{W}_1 : \mathbb{F}_s^m \to \mathbb{F}_s^w$, $\boldsymbol{W}_2 : \mathbb{F}_s^w \to \mathbb{F}_s^m$, and $\boldsymbol{b} \in \mathbb{F}_s^m$, where $w$ is the hidden dimension. Given an input $\boldsymbol{x}_i \in \mathbb{F}_s^m$, it outputs $\boldsymbol{W}_2 \mathrm{ReLU}(\boldsymbol{W}_1 \boldsymbol{x}_i + \boldsymbol{b})$, where $\mathrm{ReLU}(\boldsymbol{x}) = (\max\{0, \boldsymbol{x}_1\}, \ldots, \max\{0, \boldsymbol{x}_m\})^\top$.
5. An output function $\mathbf{OUT} : \mathbb{F}_s^m \to \mathbb{F}_s^{|\mathcal{V}|}$, parameterized as a linear transformation.

## A.3. Chain of Thought

**Definition A.2** (CoT). Let the Transformer be defined as the composition:

$$\mathrm{TF}_{\mathrm{dec}} := \mathbf{OUT} \circ (\mathrm{id} + \mathbf{FF}_L) \circ (\mathrm{id} + \mathbf{SA}_L) \circ \cdots \circ (\mathrm{id} + \mathbf{FF}_1) \circ (\mathrm{id} + \mathbf{SA}_1) \circ (\mathbf{WE} + \mathbf{PE}), \tag{3}$$

where $\mathbf{SA}_\ell$ and $\mathbf{FF}_\ell$ denote the causal attention and the feedforward layers at depth $\ell \in [L]$, respectively, and $\mathrm{id}$ denotes the identity function. The input tokens are first embedded via the word embedding function $\mathbf{WE}$ and the positional encoding $\mathbf{PE}$, and the final output is produced by a linear projection $\mathbf{OUT}$. Given an input sequence $x = (x_1, \ldots, x_n) \in \mathcal{V}^n$, we define the initial sequence as: $f_{\mathrm{cot}}^0(x) := x$. Then, the *CoT* computes recursively as:

$$f_{\mathrm{cot}}^{k+1}(x) := f_{\mathrm{cot}}^k(x) \cdot \mathrm{Dec}\left(\mathrm{TF}_{\mathrm{dec}}(f_{\mathrm{cot}}^k(x))\right), \tag{4}$$

where $\cdot$ denotes concatenation, and $\mathrm{Dec}(\cdot)$ is a decoding function that maps the output logits to a token in $\mathcal{V}$: in the *deterministic* model, $\mathrm{Dec}(z) := \arg\max_{i \in [|\mathcal{V}|]} z_i$; in the *stochastic* model, $\mathrm{Dec}(z) \sim \mathrm{Multinomial}(z_i / \sum_j z_j)$, assuming $z_i > 0$ for all $i$. The final output of the CoT model after $T(n)$ steps is defined as the last output length $m$ tokens of $f_{\mathrm{cot}}^{T(n)}(x)$.

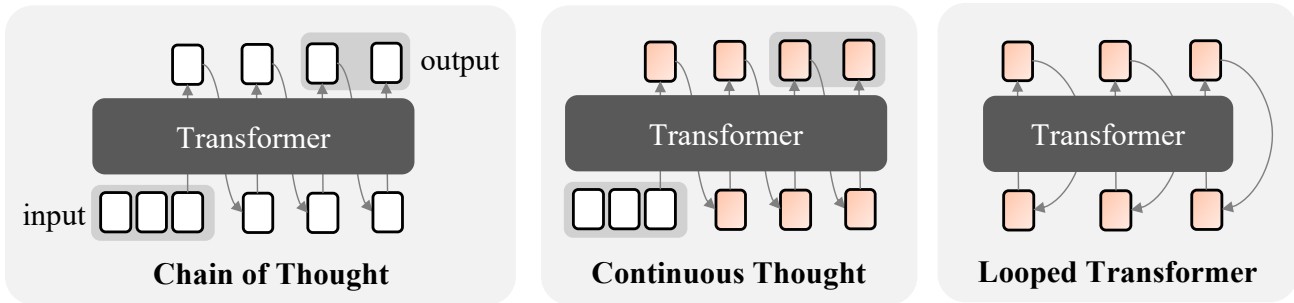

*Figure 8.* The models of reasoning paradigms based on iterative use of Transformer models.

## A.4. Continuous Thought

**Definition A.3** (Coconut). Let the Transformer block be defined as the composition:

$$\text{TF}_{\text{dec}} := (\text{id} + \mathbf{FF}_L) \circ (\text{id} + \mathbf{SA}_L) \circ \cdots \circ (\text{id} + \mathbf{FF}_1) \circ (\text{id} + \mathbf{SA}_1), \tag{5}$$

where $\mathbf{SA}_\ell$ and $\mathbf{FF}_\ell$ denote the causal attention and the feedforward layers at depth $\ell \in [L]$, respectively, and id denotes the identity function. Given an input sequence $x = (x_1, \ldots, x_n) \in \mathcal{V}^n$, we define the initial sequence as: $f_{\text{cot}}^0(x) := \mathbf{WE}(x) + \mathbf{PE}([n])$, where the input tokens are first embedded via the word embedding function $\mathbf{WE}$ and the positional encoding $\mathbf{PE}$ Then, the *continuous thought (Coconut)* computes recursively as:

$$f_{\text{ct}}^{k+1}(x) := f_{\text{ct}}^k(x) \cdot (\text{TF}_{\text{dec}}(f_{\text{cot}}^k(x) + \mathbf{PE}(k))), \tag{6}$$

where $\cdot$ denotes concatenation. The final output of the model after $T(n)$ steps is defined as the last output length $m$ tokens of $\text{Dec}\left(\mathbf{OUT}\left(f_{\text{ct}}^{T(n)}(x)\right)\right)$, where the final output is produced by a linear projection $\mathbf{OUT}$ and $\text{Dec}(\cdot)$ is a decoding function that maps the output logits to a token in $\mathcal{V}$: in the *deterministic* model, $\text{Dec}(z) := \arg\max_{i \in [|\mathcal{V}|]} z_i$; in the *stochastic* model, $\text{Dec}(z) \sim \text{Multinomial}(z_i / \sum_j z_j)$, assuming $z_i > 0$ for all $i$.

## A.5. Looped Transformer

**Definition A.4** (Looped TF). Let the Transformer block be defined as the composition:

$$\text{TF} := (\text{id} + \mathbf{FF}_L) \circ (\text{id} + \mathbf{SA}_L) \circ \cdots \circ (\text{id} + \mathbf{FF}_1) \circ (\text{id} + \mathbf{SA}_1), \tag{7}$$

where $\mathbf{SA}_\ell$ and $\mathbf{FF}_\ell$ denote the (non-causal) self-attention and feedforward layers at depth $\ell \in [L]$.

Given an input token sequence $x = (x_1, \ldots, x_n) \in \mathcal{V}^n$, the initial hidden state is: $f_{\text{loop}}^0(x) := \mathbf{WE}(x)$. At each loop iteration $k$, the hidden state is updated by:

$$f_{\text{loop}}^{k+1}(x) := \text{TF}\left(f_{\text{loop}}^k(x)\right). \tag{8}$$

The final outputs after $T(n)$ loop iterations are decoded as $\text{Dec} \circ \mathbf{OUT} \circ f_{\text{loop}}^{T(n)}(x)$, and the model's prediction is defined as the last output length $m \leq n$ tokens of this projected sequence.

## B. Deferred Proofs for Section 3

### B.1. Precision Modeling

We focus on signed floating-point numbers, following (Li et al., 2024), but omit exponents for simplicity.

**Definition B.1** (Floating-point Representation, cf. (Li et al., 2024)). Consider floating-point numbers with a mantissa part of $s$ bits and a sign bit of 1, totaling $(s + 1)$ bits. We denote the set of such floating-point numbers by $\mathbb{F}_s$, and define $B_s \triangleq \max \mathbb{F}_s$.

**Definition B.2** (Correct Rounding, cf. (Li et al., 2024)). For any $x \in \mathbb{R}$ and any closed subset $\mathbb{F} \subset \mathbb{R}$ containing 0, we define the *correct rounding* $\text{round}(x, \mathbb{F})$ as the number in $\mathbb{F}$ closest to $x$. In particular, rounding to a floating-point number with mantissa part $s$ bits is denoted by $[\cdot]_s$. Rounding applied to vectors is to be operated coordinate-wise.

They also define primitive arithmetic under finite precision by applying rounding after each basic operation. In particular, for multi-operand operations, rounding is applied after each binary operation. Finite-precision summation over more than two numbers is thus defined as follows.

**Definition B.3** (Summation with Iterative Rounding (Li et al., 2024))**.** For any $s, n \in \mathbb{N}^+$ and vector $\boldsymbol{x} \in \mathbb{R}^n$, define the *summation with iterative rounding to s-bit precision*

$$\text{sum}_s : \bigcup_{n \in \mathbb{N}^+} (\mathbb{F}_s)^n \to \mathbb{F}_s, \tag{9}$$

where, for any $n \in \mathbb{N}^+$ and $\boldsymbol{x} = (x_1, \dots, x_n) \in \mathbb{R}^n$,

$$\text{sum}_s(x) := \left[ \left[ \cdots \left[ [x_1 + x_2]_s + x_3 \right]_s + \cdots + x_{n-1} \right]_s + x_n \right]_s. \tag{10}$$

Based on this definition, all computations in the Transformer block of Definition A.1 can be represented in finite precision. The inner product and the matrix product are defined as

$$\boldsymbol{x}^\top \boldsymbol{y} := \text{sum}_s(\boldsymbol{x} \odot \boldsymbol{y}), \qquad (\boldsymbol{A}\boldsymbol{B})_{i,j} := \boldsymbol{A}_{i,:}^\top \boldsymbol{B}_{:,j}. \tag{11}$$

Throughout this section, we interpret all operations as finite-precision computations as defined above.

## B.2. Definition of Assumption 3.4

Our definition of polynomially efficient approximation follows that of Feng et al. (2023), but differs in scope: while their framework targets real-valued functions, ours applies to symbolic functions.

**Definition B.4** (Polynomially-efficient approximation)**.** We say that a function $f_n : \Sigma^{\ell(n)} \to \Sigma$ admits a *polynomially efficient approximation* if, for a sufficiently small error tolerance $0 < \delta \leq \frac{1}{3}$, there exists a feedforward network FF : $\mathbb{F}_{s(n)}^{\ell(n) \cdot |\Sigma|} \to \mathbb{F}_{s(n)}^{|\Sigma|}$, $s(n) = O(\log n)$, such that the following holds: for every input $\boldsymbol{x} = (x_1, \dots, x_{\ell(n)}) \in \Sigma^{\ell(n)}$,

$$\text{FF}(\boldsymbol{e}(x_1), \dots, \boldsymbol{e}(x_{\ell(n)}))_i = \begin{cases} \geq 1 - \delta & \text{if} \quad \boldsymbol{e}(f_n(\boldsymbol{x})) = \boldsymbol{e}_i, \\ \leq \delta & \text{else}, \end{cases} \tag{12}$$

where $\boldsymbol{e} : \Sigma \to \{0, 1\}^{|\Sigma|}$ denote the one-hot encoding. Moreover, the number of parameters of the feedforward network is bounded by a polynomial in $\ell(n)$ and $1/\delta$.

## B.3. Technical lemmas

In this section, we provide the key components for our constructive proofs.

### B.3.1. ORTHOGONAL VECTORS

We follow the notation of (Li et al., 2024). For any positive integer $s \in \mathbb{N}^+$ and $x \in \{0, 1, \dots, 2^s - 1\}$, we denote by $\text{bin}_s(x) \in \{0, 1\}^s$ the standard binary representation of $x$ using $s$ bits, defined such that $x = \sum_{i=1}^s 2^i \cdot (\text{bin}_s(x))_i$. We further define the signed binary encoding of $x$, denoted by $\text{sbin}_s(x) \in \{-1, 1\}^s$, as $\text{sbin}_s(x) = 2 \cdot \text{bin}_s(x) - (1, \dots, 1)$. Let $\boldsymbol{x}, \boldsymbol{y} \in \mathbb{R}^s$ be two vectors of the same length. We define their interleaving, denoted by $\boldsymbol{x} ^\frown \boldsymbol{y} \in \mathbb{R}^{2s}$, as follows: $(\boldsymbol{x} ^\frown \boldsymbol{y})_{2i-1} = x_i, (\boldsymbol{x} ^\frown \boldsymbol{y})_{2i} = y_i$ for all $i \in [s]$. The orthogonal vectors under finite-precision arithmetic can be:

**Lemma B.5** (Li et al., 2024)**.** *For any* $s \in \mathbb{N}^+$*, let* $\boldsymbol{q}_i = \text{sbin}_s(i) ^\frown 1_s$ *and* $\boldsymbol{k}_i = 2^{s+1} \cdot (\text{sbin}_s(i) ^\frown (-1_s))$ *for all* $i \in [2^s - 1]$*, it holds that* $\langle \boldsymbol{q}_i, \boldsymbol{k}_j \rangle_s = -B_s$ *if* $i \neq j$ *and* $\langle \boldsymbol{q}_i, \boldsymbol{k}_j \rangle_s = 0$ *if* $i = j$*. Since* $\left[ \exp(-B_s) \right]_s \leq \left[ 2^{-s-1} \right]_s = 0$*, it follows that* $\left[ \exp(\langle \boldsymbol{q}_i, \boldsymbol{k}_j \rangle_s) \right]_s = \mathbf{1}[i = j]$ *for all* $i, j \in [2^s - 1]$*.*

### B.3.2. POSITION SELECTOR

**Lemma B.6.** *For any* $m \in \mathbb{N}^+$ *and* $\boldsymbol{x} \in \mathbb{F}_s^m$ *with* $x_i > 0$ *for all* $i \in [m]$*, there exists a feedforward layer* FF : $\mathbb{F}_s^{2m} \to \mathbb{F}_s^{2m}$*, for any* $i \in [m]$*, such that*

$$(\text{id} + \text{FF})((\boldsymbol{x}, \boldsymbol{e}_i)) = (\boldsymbol{x} \odot \boldsymbol{e}_i, \boldsymbol{e}_i). \tag{13}$$

*Proof.* Let the input be $\boldsymbol{z} = (\boldsymbol{x}, \boldsymbol{e}_i) \in \mathbb{F}_s^{2m}$. Set the weight $\boldsymbol{W}_1 \in \mathbb{F}_s^{m \times 2m}$ and bias $\boldsymbol{b} \in mathbbF_s^m$ by

$$\boldsymbol{W}_1 = \begin{bmatrix} \boldsymbol{I}_m & -B_s\boldsymbol{I}_m \end{bmatrix}, \qquad \boldsymbol{b} = \boldsymbol{0}, \tag{14}$$

to have $\boldsymbol{W}_1\boldsymbol{z} + \boldsymbol{b} = \boldsymbol{x} - B_s\boldsymbol{e}_i$. Applying the ReLU activation coordinate-wise gives

$$\mathrm{ReLU}(\boldsymbol{x} - B_s\boldsymbol{e}_i)_j = \begin{cases} 0, & j = i, \\ x_j, & j \neq i. \end{cases} \tag{15}$$

Hence,

$$\boldsymbol{h} := \mathrm{ReLU}(\boldsymbol{W}_1\boldsymbol{z} + \boldsymbol{b}) = \boldsymbol{x} \odot (\boldsymbol{1} - \boldsymbol{e}_i). \tag{16}$$

Next, set the second linear layer $\boldsymbol{W}_2 \in \mathbb{F}_s^{2m \times m}$ by

$$\boldsymbol{W}_2 = \begin{bmatrix} -\boldsymbol{I}_m \\ \boldsymbol{0}_{m \times m} \end{bmatrix}. \tag{17}$$

Thus we have

$$\boldsymbol{z} + \boldsymbol{W}_2\boldsymbol{h} = \begin{bmatrix} \boldsymbol{x} \\ \boldsymbol{e}_i \end{bmatrix} + \begin{bmatrix} -\boldsymbol{h} \\ \boldsymbol{0} \end{bmatrix} = \begin{bmatrix} \boldsymbol{x} \\ \boldsymbol{e}_i \end{bmatrix} + \begin{bmatrix} -\boldsymbol{x} \odot (\boldsymbol{1} - \boldsymbol{e}_i) \\ \boldsymbol{0} \end{bmatrix} = \begin{bmatrix} \boldsymbol{x} \odot \boldsymbol{e}_i \\ \boldsymbol{e}_i \end{bmatrix}. \tag{18}$$

$\square$

### B.3.3. FEEDFORWARD LAYERS

**Lemma B.7.** *Let $N \in \mathbb{N}^+$. For each $i \in [N]$, let $f_i : \Sigma^{\ell_i} \to \Sigma$ admit polynomially-efficient approximation by $\mathrm{FF}_i$. Then there exists a feedforward layer $\mathrm{FF} : \mathbb{F}_s^{\sum_{i=1}^N \ell_i |\Sigma|} \to \mathbb{F}_s^{N|\Sigma|}$ such that for every input tuple $\boldsymbol{x}_i = (x_1^{(i)}, \ldots, x_{\ell_i}^{(i)}) \in \Sigma^{\ell_i}$,*

$$\mathrm{FF}\left(\Big\|_{i=1}^N \big(\boldsymbol{e}(x_1^{(i)}), \ldots, \boldsymbol{e}(x_{\ell_i}^{(i)})\big)\right) = \Big\|_{i=1}^N \mathrm{FF}_i(\boldsymbol{x}_i), \tag{19}$$

*where $\|$ denotes concatenation.*

*Proof.* For each $i \in [N]$, by Definition B.4, there exist width $w_i \in \mathbb{N}$ and parameters

$$\boldsymbol{W}_1^{(i)} \in \mathbb{F}_s^{w_i \times \ell_i |\Sigma|}, \quad \boldsymbol{W}_2^{(i)} \in \mathbb{F}_s^{|\Sigma| \times w_i}, \quad \boldsymbol{b}^{(i)} \in \mathbb{F}_s^{w_i} \tag{20}$$

such that $\mathrm{FF}_i(\boldsymbol{x}_i) = \boldsymbol{W}_2^{(i)}\mathrm{ReLU}(\boldsymbol{W}_1^{(i)}\boldsymbol{e}(\boldsymbol{x}_i) + \boldsymbol{b}^{(i)})$.

Now, define block-diagonal matrices

$$\boldsymbol{W}_1 := \bigoplus_{i=1}^N \boldsymbol{W}_1^{(i)}, \qquad \boldsymbol{W}_2 := \bigoplus_{i=1}^N \boldsymbol{W}_2^{(i)}, \qquad \boldsymbol{b} := \bigoplus_{i=1}^N \boldsymbol{b}^{(i)}. \tag{21}$$

Then the single feedforward layer

$$\mathrm{FF}(\boldsymbol{x}) := \boldsymbol{W}_2\mathrm{ReLU}(\boldsymbol{W}_1\boldsymbol{x} + \boldsymbol{b}) \tag{22}$$

applies each block independently to its corresponding input, yielding exactly $\mathrm{FF}(\boldsymbol{x}) = \big\|_{i=1}^N \mathrm{FF}_i(\boldsymbol{x}_i)$. $\square$

**Lemma B.8.** *Let $f : \Sigma^\ell \to \Sigma$ be a function that admits polynomially-efficient approximation (Definition B.4). Then, there exist two feedforward layers*

$$\mathrm{FF}_1 : \mathbb{F}_s^{(1+\ell)|\Sigma|+1} \to \mathbb{F}_s^{(1+\ell)|\Sigma|+1}, \quad \mathrm{FF}_2 : \mathbb{F}_s^{(1+\ell)|\Sigma|+1} \to \mathbb{F}_s^{(1+\ell)|\Sigma|+1}, \tag{23}$$

*such that, for every $\boldsymbol{x} = (x_1, \ldots, x_\ell) \in \Sigma^\ell$ and $t \in \{0, 1\}$,*

$$(\mathrm{id} + \mathrm{FF}_2) \circ (\mathrm{id} + \mathrm{FF}_1)\big(\boldsymbol{0}, \boldsymbol{e}(x_1), \ldots, \boldsymbol{e}(x_\ell), t\big) = (t \cdot \boldsymbol{e}(f(\boldsymbol{x})), \boldsymbol{e}(x_1), \ldots, \boldsymbol{e}(x_\ell), t). \tag{24}$$

*Proof.* Let $n := |\Sigma|$ and $L := \ell n$. Write $\boldsymbol{u} = (\boldsymbol{e}(x_1), \ldots, \boldsymbol{e}(x_\ell)) \in \{0, 1\}^L$. By Definition B.4, there exist $w_f \in \mathbb{N}$, matrices $\boldsymbol{W}_1^{(f)} \in \mathbb{F}_s^{w_f \times L}$, $\boldsymbol{W}_2^{(f)} \in \mathbb{F}_s^{n \times w_f}$, and bias $\boldsymbol{b}^{(f)} \in \mathbb{F}_s^{w_f}$ such that

$$\mathrm{FF}_f(\boldsymbol{u}) := \boldsymbol{W}_2^{(f)} \mathrm{ReLU}(\boldsymbol{W}_1^{(f)} \boldsymbol{u} + \boldsymbol{b}^{(f)}), \quad \mathrm{FF}_f(\boldsymbol{u})_i = \begin{cases} \geq 1 - \delta & \text{if } \boldsymbol{e}(f(\boldsymbol{x})) = \boldsymbol{e}_i, \\ \leq \delta & \text{otherwise.} \end{cases} \tag{25}$$

Set the first layer as

$$\boldsymbol{W}_1^{(1)} = \begin{bmatrix} \boldsymbol{0} & \boldsymbol{W}_1^{(f)} & \boldsymbol{0}_{w_f \times 1} \\ \boldsymbol{0} & \boldsymbol{I}_L & \boldsymbol{0}_{L \times 1} \end{bmatrix}, \quad \boldsymbol{b}^{(1)} = \begin{bmatrix} \boldsymbol{b}^{(f)} \\ \boldsymbol{0}_L \end{bmatrix}, \quad \boldsymbol{W}_2^{(1)} = \begin{bmatrix} \boldsymbol{W}_2^{(f)} & \boldsymbol{0} \\ \boldsymbol{0}_{L \times w_f} & \boldsymbol{0} \\ \boldsymbol{0}_{1 \times w_f} & \boldsymbol{0} \end{bmatrix}, \tag{26}$$

and define $\mathrm{FF}_1(\boldsymbol{0}, \boldsymbol{u}, t) := \boldsymbol{W}_2^{(1)} \mathrm{ReLU}(\boldsymbol{W}_1^{(1)}(\boldsymbol{0}, \boldsymbol{u}, t) + \boldsymbol{b}^{(1)})$. Then, it holds that

$$\mathrm{FF}_1(\boldsymbol{0}, \boldsymbol{u}, t) = \boldsymbol{W}_2^{(1)} \mathrm{ReLU}\left(\begin{bmatrix} \boldsymbol{W}_1^{(f)} \boldsymbol{u} + \boldsymbol{b}^{(f)} \\ \boldsymbol{u} \end{bmatrix}\right) \tag{27}$$

$$= \begin{bmatrix} \boldsymbol{W}_2^{(f)} \mathrm{ReLU}(\boldsymbol{W}_1^{(f)} \boldsymbol{u} + \boldsymbol{b}^{(f)}) \\ \boldsymbol{0} \end{bmatrix} \tag{28}$$

$$= \begin{bmatrix} \mathrm{FF}_f(\boldsymbol{u}) \\ \boldsymbol{0} \end{bmatrix}. \tag{29}$$

Thus we have $(\mathrm{id} + \mathrm{FF}_1)(\boldsymbol{0}, \boldsymbol{u}, t) = (\mathrm{FF}_f(\boldsymbol{u}), \boldsymbol{u}, t)$.

For the second layer, choose $\delta \leq \frac{1}{3}$ and $M \geq 1$ and set

$$\boldsymbol{W}_1^{(2)} = \begin{bmatrix} 2\boldsymbol{I}_n & \boldsymbol{0}_{n \times L} & M\boldsymbol{1}_n \\ 2\boldsymbol{I}_n & \boldsymbol{0}_{n \times L} & M\boldsymbol{1}_n \\ \boldsymbol{I}_n & \boldsymbol{0}_{n \times L} & \boldsymbol{0}_{n \times 1} \\ -\boldsymbol{I}_n & \boldsymbol{0}_{n \times L} & \boldsymbol{0}_{n \times 1} \end{bmatrix}, \quad \boldsymbol{b}^{(2)} = \begin{bmatrix} -M\boldsymbol{1}_n \\ (1 - M)\boldsymbol{1}_n \\ \boldsymbol{0}_n \\ \boldsymbol{0}_n \end{bmatrix}, \quad \boldsymbol{W}_2^{(2)} = \begin{bmatrix} \boldsymbol{I}_n & -\boldsymbol{I}_n & -\boldsymbol{I}_n & \boldsymbol{I}_n \\ \boldsymbol{0}_{L \times n} & \boldsymbol{0}_{L \times n} & \boldsymbol{0}_{L \times n} & \boldsymbol{0}_{L \times n} \\ \boldsymbol{0}_{1 \times n} & \boldsymbol{0}_{1 \times n} & \boldsymbol{0}_{1 \times n} & \boldsymbol{0}_{1 \times n} \end{bmatrix}. \tag{30}$$

and define $\mathrm{FF}_2(\boldsymbol{y}) := \boldsymbol{W}_2^{(2)} \mathrm{ReLU}(\boldsymbol{W}_1^{(2)} \boldsymbol{y} + \boldsymbol{b}^{(2)})$. Then it holds that, for $\boldsymbol{z} := \mathrm{FF}_f(\boldsymbol{u})$,

$$\mathrm{FF}_2(\boldsymbol{z}, \boldsymbol{u}, t) = \boldsymbol{W}_2^{(2)} \begin{bmatrix} \mathrm{ReLU}(2\boldsymbol{z} + Mt\boldsymbol{1}_n - M\boldsymbol{1}_n) \\ \mathrm{ReLU}(2\boldsymbol{z} + Mt\boldsymbol{1}_n + (1 - M)\boldsymbol{1}_n) \\ \mathrm{ReLU}(\boldsymbol{z}) \\ \mathrm{ReLU}(-\boldsymbol{z}) \end{bmatrix} \tag{31}$$

$$= \boldsymbol{W}_2^{(2)} \begin{bmatrix} \mathrm{ReLU}(2\boldsymbol{z} + M(t - 1)\boldsymbol{1}_n) \\ \mathrm{ReLU}(2\boldsymbol{z} + 1 + M(t - 1)\boldsymbol{1}_n) \\ \boldsymbol{z} \\ \boldsymbol{0} \end{bmatrix} \tag{32}$$

$$= \begin{bmatrix} \mathrm{ReLU}(\boldsymbol{z} - \delta + M(t - 1)) - \mathrm{ReLU}(\boldsymbol{z} - 1 + M(t - 1)) - \boldsymbol{z} \\ \boldsymbol{0} \\ \boldsymbol{0} \end{bmatrix}, \tag{33}$$

where it satisfies that

$$\mathrm{ReLU}(\boldsymbol{z} - \delta + M(t - 1)) - \mathrm{ReLU}(\boldsymbol{z} - \delta - 1 + M(t - 1)) = \begin{cases} \boldsymbol{e}(f(\boldsymbol{x})) & \text{if } t = 1, \\ \boldsymbol{0} & \text{if } t = 0. \end{cases} \tag{34}$$

Therefore, the composition satisfies $(\mathrm{id} + \mathrm{FF}_2) \circ (\mathrm{id} + \mathrm{FF}_1)(\boldsymbol{0}, \boldsymbol{u}, t) = (t \cdot \boldsymbol{e}(f(\boldsymbol{x})), \boldsymbol{u}, t)$. $\square$

**B.4. Proof for Theorem 3.5**

*Proof.* Let $G_n = (V_n, E_n)$ be a computation graph, where $\mathcal{F} = \{f_1, f_2, \ldots, f_{|\mathcal{F}|}\}$. Each node $v \in V_n$ is labeled by a one-hot vector $\boldsymbol{e}(v) \in \{0,1\}^{|\mathcal{F}|}$ indicating the function assigned to $v$ from the finite set $\mathcal{F}$. Let $v_1, v_2, \ldots, v_{|V_n|}$ denote a fixed topological ordering of $V_n$, with inputs appearing first and outputs last. For each function $f_i \in \mathcal{F}$, let $C_{f_i}(n) := \max\{|\mathrm{pred}(v)| : v \in V_n, \boldsymbol{e}(v) = \boldsymbol{e}_i\}$. and define $C_{\mathrm{sum}}(n) := \sum_{f \in \mathcal{F}} C_f(n)$, $C_{\max}(n) := \max_{f \in \mathcal{F}} C_f(n)$.

Let the precision be $s(n) = C \cdot \lceil \log_2 n \rceil$ where $C \in \mathbb{N}$ is a sufficiently large integer such that $2^{s(n)} \geq n^C$ exceeds the maximum polynomial step bound under consideration. We denote by $\mathrm{pred}(v_i) \in \mathbb{F}_{s(n)}^{C_{\max}(n)}$ the vector of predecessor indices of node $v_i$; that is, if $v_i$ has $d \leq C_{\max}(n)$ incoming edges from nodes $v_{j_1}, \ldots, v_{j_d}$, then $\mathrm{pred}(v_i) = (j_1, \ldots, j_d, \boldsymbol{0})$, where zeros are used for padding so that the length is exactly $C_{\max}(n)$.

Let the vocabulary be $\mathcal{V} = \Sigma$. At decoding step $k$, the model has access to the concatenated sequence

$$(x_1, x_2, \ldots, x_n, y_1, y_2, \ldots, y_k) \in \Sigma^{n+k}, \tag{35}$$

where $x = (x_1, \ldots, x_n) \in \Sigma^n$ denotes the input, and $y_i$ is the token generated at the $i$-th CoT step. For each node $v_j$, let $v_j(x)$ denote its value on input $x$. We assume that every intermediate output satisfies $y_i = v_{n+i}(x)$. Under this assumption, we prove by induction that the model generates the next token correctly, i.e., $y_{k+1} = v_{n+k+1}(x)$.

**Embedding** The embedding at position $i \in [n+k]$, denoted by $\boldsymbol{h}_i^{(0)} \in \mathbb{F}_{s(n)}^m$, where $m := |\Sigma| + |\mathcal{F}| + (1 + C_{\max}(n))s(n) + |\Sigma|C_{\mathrm{sum}}(n)$ is defined as

$$\boldsymbol{h}_i^{(0)} = \Big(\boldsymbol{e}(v_i(x)), \boldsymbol{e}(v_{i+1}), \mathbf{sbin}_{s(n)}(i), \mathbf{sbinpred}_{s(n)}(v_{i+1}), \boldsymbol{0}_{|\Sigma|C_{\mathrm{sum}}(n)}\Big), \tag{36}$$

where $\boldsymbol{e} \colon \Sigma \to \{0,1\}^{|\Sigma|}$ denote the one-hot encoding of the symbol and, $\mathbf{sbinpred}_{s(n)}(v) \in \mathbb{F}_{s(n)}^{C_{\max}(n) \cdot s(n)}$ encodes the binary representations of the predecessor indices:

$$\mathbf{sbinpred}_{s(n)}(v_i) := \big(\mathsf{sbin}_{s(n)}(\mathrm{pred}(v_i)_0), \ldots, \mathsf{sbin}_{s(n)}(\mathrm{pred}(v_i)_{C_{\max}(n)})\big). \tag{37}$$

This embedding is constructed, for $z \in \Sigma$, as

$$\mathbf{WE}(z) = (\boldsymbol{e}(z), \boldsymbol{0}), \quad \mathbf{PE}(i) = \Big(\boldsymbol{0}, \boldsymbol{e}(v_{i+1}), \mathsf{sbin}_{s(n)}(i), \mathbf{sbinpred}_{s(n)}(v_{i+1}), \boldsymbol{0}\Big). \tag{38}$$

**Attention layer** The first attention layer consists of $C_{\max(n)}$ heads. The $h$-th head is configured to attend to the position corresponding to the $h$-th predecessor. Specifically, for each position $i$ and head $h \in [C_{\max(n)}]$, the attention vectors are defined as:

$$\boldsymbol{q}_{i,h} = \mathsf{sbin}_{s(n)}(\mathrm{pred}(v_{i+1})_h) \frown 1_{s(n)}, \tag{39}$$

$$\boldsymbol{k}_{i,h} = 2^{s(n)+1} \cdot \mathsf{sbin}_{s(n)}(i) \frown (-1_{s(n)}), \tag{40}$$

$$\mathbf{v}_{i,h} = \boldsymbol{e}(v_i(x)), \tag{41}$$

where vectors of different lengths are zero-padded to match the dimension. By Lemma B.5, each attention head of the last position $i = n + k$ retrieves the predecessor's value of $v_{n+k}$

$$\boldsymbol{a}_{n+k,h} = \boldsymbol{e}\big(v_{\mathrm{pred}(v_{n+k+1})_h}(x)\big) \tag{42}$$

With an appropriate output projection $\boldsymbol{O}$ such that

$$\boldsymbol{O}(\boldsymbol{a}_{i,1}, \ldots, \boldsymbol{a}_{i,H}) = (\boldsymbol{0}, \boldsymbol{e}(v_{\mathrm{pred}(v_{n+k+1})_0}(x)), \ldots, \boldsymbol{e}(v_{\mathrm{pred}(v_{n+k+1})_{C_{\max(n)}}}(x)), \boldsymbol{0}), \tag{43}$$

the hidden state at position $n + k$ after the attention layer is given by

$$\boldsymbol{h}_{n+k}^{(0.5)} = \Big(\boldsymbol{e}(v_{n+k}(x)), \boldsymbol{e}(v_{n+k+1}), \mathsf{sbin}_{s(n)}(n+k), \mathbf{sbinpred}_{s(n)}(v_{n+k+1}), \tag{44}$$

$$\underbrace{\boldsymbol{e}(v_{\mathrm{pred}(v_{n+k+1})_0}(x)), \ldots, \boldsymbol{e}(v_{\mathrm{pred}(v_{n+k+1})_{C_{\max(n)}}}(x))}_{\text{updated}}, \boldsymbol{0}_{C_{\mathrm{sum}}(n)|\Sigma|}\Big). \tag{45}$$

The second and third attention layers are disabled (i.e., all attention weights are set to zero).

**Feed-forward layer** By Lemma B.7, a single feed-forward layer can approximate multiple functions by partitioning the input into blocks. The first feed-forward layer then places the arguments, gathered by attention, into the correct positions. By Lemma B.6, where the vector $e_i$ therein corresponds to $\mathbf{1}_{|\Sigma|}$ here, the hidden state at the last position, denoted by $h_{n+k}^{(1)}$, becomes

$$\left((h_{n+k}^{(0.5)})_{1:r}, \Big\|_{j=1}^{|\mathcal{F}|} \big(e(v_{\text{pred}(v_{n+k+1})_1}(x)) \cdot 1, \ldots, e(v_{\text{pred}(v_{n+k+1})_{C_j(n)}}(x)) \cdot 1\big)\right), \tag{46}$$

where $r = |\Sigma| + |\mathcal{F}| + (1 + C_{\max})(n)s(n)$.

By Assumption 3.4 and Lemmas B.7 and B.8, there exist feed-forward layers $\text{FF}_2, \text{FF}_3 : \mathbb{F}_{s(n)}^m \to \mathbb{F}_{s(n)}^m$ such that, for every input tuple $x_j := (x_1^{(j)}, \ldots, x_{C_j(n)}^{(j)}) \in \Sigma^{C_j(n)}$ for $j \in [|\mathcal{F}|]$ and every $t \in \{0,1\}^{|\mathcal{F}|}$, the composition $\mathcal{FF} := (\text{id} + \text{FF}_3) \circ (\text{id} + \text{FF}_2)$ satisfies

$$\mathcal{FF}\left(*, t, *, \Big\|_{j=1}^{|\mathcal{F}|} (e(x_1^{(j)}), \ldots, e(x_{C_j(n)}^{(j)}))\right) = \left(\sum_j t_j \cdot e(f_j(x_j)), *\right), \tag{47}$$

zwhere $*$ denotes an unspecified vector. Since the second and third attention layers are disabled, after the third layer, applying the second and third feed-forward layers $\text{FF}_2, \text{FF}_3$, the hidden state becomes

$$h_{n+k}^{(3)} = \mathcal{FF}\left(h_{n+k}^{(1)}\right) \tag{48}$$

$$= \mathcal{FF}\left(*, e(v_{n+k+1}), *, \Big\|_{j=1}^{|\mathcal{F}|} \big(e(v_{\text{pred}(v_{n+k+1})_1}(x)), \ldots, e(v_{\text{pred}(v_{n+k+1})_{C_j(n)}}(x))\big)\right) \tag{49}$$

$$= \left(\sum_{j=1}^{|\mathcal{F}|} e(v_{n+k+1})_j \cdot e\Big(f_j\big(e(v_{\text{pred}(v_{n+k+1})_1}(x)), \ldots, e(v_{\text{pred}(v_{n+k+1})_{C_j(n)}}(x))\big)\Big), *\right) \tag{50}$$

$$= \left(e\Big(f_l\big(e(v_{\text{pred}(v_{n+k+1})_1}(x)), \ldots, e(v_{\text{pred}(v_{n+k+1})_{C_l(n)}}(x))\big)\Big), *\right), \quad \text{where} \quad e(v_{n+k+1}) = e_l \tag{51}$$

$$= \left(e\big(v_{n+k+1}(x)\big), *\right). \tag{52}$$

**Output layer** The final output is given by

$$h_{n+k} = \mathbf{OUT}(h_{n+k}^{(3)}) = \begin{bmatrix} I_{|\Sigma|} & 0 \end{bmatrix} h_{n+k}^{(3)} = e(v_{n+k+1}(x)). \tag{53}$$

The decoding function then outputs the symbol corresponding to the maximum score,

$$y_{n+k+1} = \text{Dec}(h_{n+k}) = \arg\max_{j \in [|\Sigma|]} e(v_{n+k+1}(x)) = v_{n+k+1}(x). \tag{54}$$

By induction on $k$, the model computes the values at all nodes in topological order. The parameter size of the model is determined by the requirements of the feedforward layers, $O(\text{ff\_param}(G_n))$. While the dimensions and heads of the attention layers depend on $C_{\max}(n)$, which is precisely what is already required for the feedforward layers to approximate the target functions. $\square$

### B.5. Proof of Theorem 3.6 for Looped Transformer

*Proof.* In the proof, we assume that the computation graph contains at most $n$ output nodes. This assumption is without loss of generality: if the number of output nodes exceeds the number of input nodes, we can simply pad the input with dummy nodes (e.g., fixed zeros), thereby reducing the setting to the same case.

We construct a model in which (1) the attention layer aggregates the inputs, and (2) the looped feed-forward layer performs the computation of all nodes in parallel, as illustrated in Figure 9. We first show that a feedforward layer followed by an attention layer can copy all input tokens to each position. Assume, given an input sequence $x = (x_1, x_2, \ldots, x_n) \in \Sigma^n$ and one-hot encoding $e : \Sigma \to \{0,1\}^{|\Sigma|}$. Assume each input at position $i \in [n]$ is embedded as

$$h_i = \big(\mathbf{0}, \ e(x_i), \ e_i\big) \in \{0,1\}^{n|\Sigma|+|\Sigma|+n}, \tag{55}$$

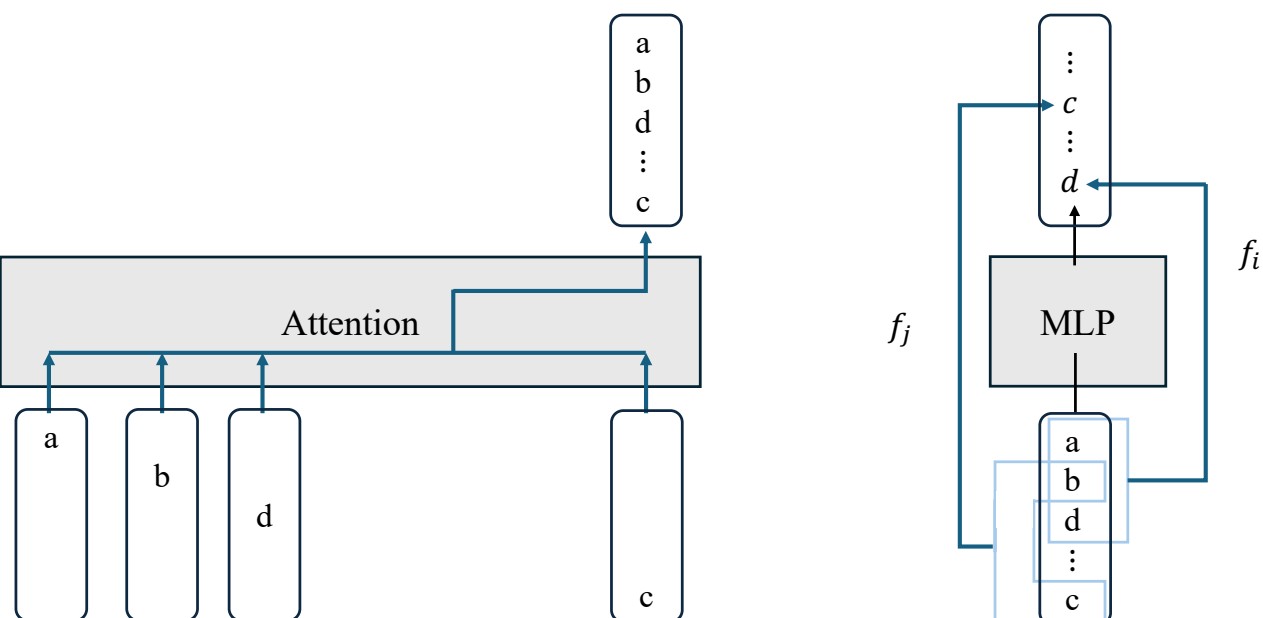

*Figure 9.* Illustration of the role of the attention and feedforward layers in looped TFs for evaluating DAGs: the attention layer uniformly attends to and aggregates all inputs at each position, while the feedforward layer simultaneously simulates the functions of the nodes.

where $e_i \in \{0, 1\}^n$. By Lemma B.6, when substituting $\boldsymbol{x} = (\boldsymbol{e}(x_1), \dots, \boldsymbol{e}(x_n))$ into the lemma, there exists a feed-forward layer $\mathrm{FF}_1$ such that

$$(\mathrm{id} + \mathrm{FF}_1)(\boldsymbol{h}_i) = \big( (\boldsymbol{e}_i)_1 \cdot \boldsymbol{e}(x_i),\ (\boldsymbol{e}_i)_2 \cdot \boldsymbol{e}(x_i),\ \dots,\ (\boldsymbol{e}_i)_n \cdot \boldsymbol{e}(x_i),\ \boldsymbol{e}(x_i),\ \boldsymbol{e}_i \big). \tag{56}$$

To aggregate all positions via uniform attention, we use a single-head attention layer with:

$$\mathbf{q}_i = \boldsymbol{k}_i = \mathbf{1}_{n|\Sigma|}, \quad \mathbf{v}_i = n\big((\boldsymbol{e}_i)_1 \cdot \boldsymbol{e}(x_i),\ (\boldsymbol{e}_i)_2 \cdot \boldsymbol{e}(x_i),\ \dots,\ (\boldsymbol{e}_i)_n \cdot \boldsymbol{e}(x_i)\big) \quad \text{for all } i \in [n], \tag{57}$$

with an appropriate output projection, the output of the attention layer, at position $i$, becomes

$$\frac{1}{n} \sum_{j=1}^{n} 1 \cdot n\boldsymbol{h}_j \;=\; \Big( \sum_{j=1}^{n} (\boldsymbol{e}_j)_1\, \boldsymbol{e}(x_j),\ \sum_{j=1}^{n} (\boldsymbol{e}_j)_2\, \boldsymbol{e}(x_j),\ \dots,\ \sum_{j=1}^{n} (\boldsymbol{e}_j)_n\, \boldsymbol{e}(x_j) \Big) \tag{58}$$

$$= \big( \boldsymbol{e}(x_1),\ \boldsymbol{e}(x_2),\ \dots,\ \boldsymbol{e}(x_n) \big). \tag{59}$$

Then, we show that the feed-forward layer can encode the entire computation graph into its weights and simulate all nodes simultaneously. Let the flag vector for each node be $(t_1, \dots, t_N) \in \{0, 1\}^N$, where $N := |V_n| = \mathrm{size}(G_n)$. By Lemmas B.7 and B.8, there exist feed-forward layers $\mathrm{FF}_2, \mathrm{FF}_3 : \mathbb{F}_s^{N(|\Sigma|+1)} \to \mathbb{F}_s^{N(|\Sigma|+1)}$ such that, for the input vector $(z_1, \dots, z_N) \in \Sigma^N.$,

$$\mathcal{FF}\Big( \boldsymbol{e}(z_1), t_1, \dots, \boldsymbol{e}(z_N(x)), t_N \Big) = \Big\|_{i=1}^{N} \Big( t_i \cdot \boldsymbol{e}\big(f_{v_i}(\boldsymbol{z}^{(i)})\big),\ \frac{1}{m_i} \sum_{j=1}^{m_i} t_{p_{i,j}} \Big), \tag{60}$$

where $\mathcal{FF} := (\mathrm{id} + \mathrm{FF}_3) \circ (\mathrm{id} + \mathrm{FF}_2)$. Here, $f_{v_i}$ denotes the function associated with node $v_i$, and $p_{i,1}, \dots, p_{i,m_i}$ denote the indices of the predecessor nodes of $v_i$, and $\boldsymbol{z}^{(i)} = (z_{p_{i,1}}, \dots, z_{p_{i,m_i}})$ denotes their values. The last term $\frac{1}{m_i} \sum_{j=1}^{m_i} t_{p_{i,j}}$ can be obtained using a linear layer.

For the $k$-th loop, assume by induction that the hidden state is

$$\boldsymbol{h}(k) := \big( t_{1,k-1} \cdot \boldsymbol{e}(v_1(x)), t_{1,k}, t_{2,k-1} \cdot \boldsymbol{e}(v_2(x)), t_{2,k}, \dots, t_{N,k-1} \cdot \boldsymbol{e}(v_N(x)), t_{N,k} \big), \tag{61}$$

where $v_i(x) \in \Sigma$ denotes the value computed by node $v_i$ given the input $x$, and $t_{i,k} \in \{0, 1\}$ indicates whether node $v_i$ lies within depth at most $k$. Under this assumption, it holds that

$$\mathcal{FF}(\boldsymbol{h}(k)) = \Big\|_{i=1}^{N} \Big( t_{i,k} \cdot \boldsymbol{e}\big(f_{v_i}(v_{p_{i,1}}(x), v_{p_{i,2}}(x), \dots, v_{p_{i,m_i}}(x))\big), \frac{1}{m_i} \sum_{j=1}^{m_i} t_{(p_{i,j},k)} \Big), \tag{62}$$

$$= \Big\|_{i=1}^{N} \Big( t_{i,k} \cdot \boldsymbol{e}\big(v_i(x)\big)\big), \ t_{i,k+1} \Big) = \boldsymbol{h}(k+1). \tag{63}$$

To extract the output node corresponding to each position denoted by $o_i$, in the final loop iteration, by Lemma B.6, there exists a feedforward layer $\mathrm{FF}_4$ such that

$$(\mathrm{id} + \mathrm{FF}_4)(\boldsymbol{h}(k), \boldsymbol{e}_{o_i}) = \big( t_{o_i,k} \cdot \boldsymbol{e}(v_{o_i}(x)), \ * \big). \tag{64}$$

**Summary** We construct the looped model as follows. Each input token $x_i \in \Sigma$ at position $i \in [n]$ is embedded as

$$\boldsymbol{h}_i^{(0)} = \big( \boldsymbol{e}_i, \ \boldsymbol{e}_{o_i}, \ \boldsymbol{e}(x_i), \ 0, \ \boldsymbol{e}(x_i), \ 0, \ \dots, \ \boldsymbol{e}(x_i), \ 0, \ \boldsymbol{0}_{(1+|\Sigma|)(N-n)+|\Sigma|} \big) \in \{0,1\}^{2n+(1+|\Sigma|)N+|\Sigma|}. \tag{65}$$

The first attention layer is an identity map, while the first feedforward layers compute

$$\boldsymbol{h}_i^{(1)} = \big( \boldsymbol{e}_i, \ \boldsymbol{e}_{o_i}, \ (\boldsymbol{e}_i)_1 \cdot \boldsymbol{e}(x_i), \ 0, \ (\boldsymbol{e}_i)_2 \cdot \boldsymbol{e}(x_i), \ 0, \ \dots, \ (\boldsymbol{e}_i)_n \cdot \boldsymbol{e}(x_i), \ 0, \ \boldsymbol{0}_{(1+|\Sigma|)(N-n)+|\Sigma|} \big). \tag{66}$$

The second attention layer uniformly gathers all positions and appends a constant $1$ to each block:

$$\boldsymbol{h}_i^{(1.5)} = \big( \boldsymbol{e}_i, \ \boldsymbol{e}_{o_i}, \ \boldsymbol{e}(x_1), \ 1, \ \boldsymbol{e}(x_2), \ 1, \ \dots, \ \boldsymbol{e}(x_n), \ 1, \ \boldsymbol{0}_{(1+|\Sigma|)(N-n)+|\Sigma|} \big) \tag{67}$$

$$= \big( \boldsymbol{e}_i, \ \boldsymbol{e}_{o_i}, \ \boldsymbol{h}(0), \ \boldsymbol{0}_{|\Sigma|} \big). \tag{68}$$

We now proceed by induction. Assume that at the $k$-th iteration, the output after the second attention layer at position $i$ is

$$\boldsymbol{h}_{k,i}^{(1.5)} = \big( \boldsymbol{e}_i, \ \boldsymbol{e}_{o_i}, \ \boldsymbol{h}(k), \ t_{o_i,k-1} \cdot \boldsymbol{e}(v_{o_i}(x)) \big), \tag{69}$$

where $t_{o_i,k-1} := \big( \boldsymbol{h}_{k-1,i}^{(1.5)} \big)_{2n+(1+|\Sigma|)(o_i-1)}$ denotes the indicator flag showing whether the $o_i$-th node has been reached, i.e., whether it lies at depth $k-1$.

After passing through the second feedforward layer, the third attention layer with all weights set to zero, and the third feedforward layer, the hidden state updates to

$$\boldsymbol{h}_{k,i}^{(3)} = \big( \boldsymbol{e}_i, \ \boldsymbol{e}_{o_i}, \ \boldsymbol{h}(k+1), \ t_{o_i,k-1} \cdot \boldsymbol{e}(v_{o_i}(x)) \big). \tag{70}$$

After the fourth feedforward layer, the hidden state becomes

$$\boldsymbol{h}_{k,i}^{(4)} = \big( \boldsymbol{e}_i, \ \boldsymbol{e}_{o_i}, \ \boldsymbol{h}(k+1), \ t_{o_i,k} \cdot \boldsymbol{e}(v_{o_i}(x)) \big). \tag{71}$$

By induction, after the final iteration of depth $\mathrm{depth}(G_n)$ of the computation graph $G_n$, we obtain

$$\boldsymbol{h}_{\mathrm{depth}(G_n),i}^{(4)} = \big( *, \ t_{o_i,\mathrm{depth}(G_n)} \cdot \boldsymbol{e}(v_{o_i}(x)) \big) = \big( *, \ \boldsymbol{e}(v_{o_i}(x)) \big). \tag{72}$$

The final output is given by

$$z_i = \mathbf{OUT}(\boldsymbol{h}_{\mathrm{depth}(G_n),i}^{(4)}) = \begin{bmatrix} \boldsymbol{0} & \boldsymbol{I}_{|\Sigma|} \end{bmatrix} (\boldsymbol{h}_{\mathrm{depth}(G_n),i}^{(4)}) = \boldsymbol{e}(v_{o_i}(x)). \tag{73}$$

The decoding function then selects the symbol corresponding to the maximum score:

$$y_i = \mathrm{Dec}(z_i) = \arg\max_{j \in [|\Sigma|]} \big( \boldsymbol{e}(v_{o_i}(x)) \big)_j = v_{o_i}(x). \tag{74}$$

The parameter size grows proportionally with the input dimension $\mathrm{size}(G_n)$, since the function must be approximated along each dimension. Therefore, it can be bounded by $O(\mathrm{ff\_param}(G_n) \cdot \mathrm{size}(G_n))$. $\qquad\square$

**Discussion:** Our proof compresses the entire input into a single position before applying an arbitrary feed-forward computation, which may appear to deviate from the standard Transformer architecture. An alternative approach is to distribute computation across multiple positions, as shown by (Sanford et al., 2024b), which can reduce the required embedding dimension. We nevertheless adopt the single-position construction to isolate the core characteristics of looped models: attention is used purely as an information aggregation mechanism, a single Transformer layer suffices, and all computation is carried out by the feed-forward network in latent space. This latent-space computation is strictly more expressive than computation in the language space. This is supported by recent work for looped ReLUs (Liang et al., 2024).

### B.6. Proof of Theorem 3.6 for Continuous Thought

*Proof.* The proofs are based on the construction for CoT and looped TF. Let the vocabulary be $\mathcal{V} = \Sigma$ and each node $v \in V_n$ is labeled by a one-hot vector $\boldsymbol{e}(v) \in \{0,1\}^{|\mathcal{F}|}$. At decoding step $k$, the model has access to the concatenated sequence

$$(x_1, x_2, \ldots, x_n, h_1, h_2, \ldots, h_k) \tag{75}$$

where $x = (x_1, \ldots, x_n) \in \Sigma^n$ denotes the input, and $h_i \in \mathbb{F}_s^m$ is the hidden state generated at the $i$-th Coconut step. For each node $v_j$, let $v_j(x)$ denote its value on input $x$. We assume that

$$h_k := \big(t_{1,k-1} \cdot \boldsymbol{e}(v_1(x)), \, t_{1,k}, \, t_{2,k-1} \cdot \boldsymbol{e}(v_2(x)), \, t_{2,k}, \, \ldots, \, t_{N,k-1} \cdot \boldsymbol{e}(v_N(x)), \, t_{N,k}, \, \boldsymbol{e}'_k, \, \boldsymbol{0}_{|\Sigma|}\big),$$

where $N := \text{size}(G_n)$, $t_{i,k} \in \{0,1\}$ indicates whether node $v_i$ lies within depth at most $k$, and $\boldsymbol{e} : \Sigma \to \{0,1\}^{|\Sigma|}$ denotes one-hot encoding, and the vector $\boldsymbol{e}'_k \in \{0,1\}^N$ is defined by

$$\boldsymbol{e}'_k = \begin{cases} \boldsymbol{0} & k < \text{depth}(G_n), \\ \boldsymbol{e}(v_{o_i}) & k = \text{depth}(G_n) + i, \end{cases} \tag{76}$$

where $v_{o_i}$ denotes the $i$-th output node, and $\boldsymbol{e}'_k$ can be encoded by positional embeddings. Under this assumption, we prove by induction that the model generates $h_{k+1}(x)$.

We first consider the base case $k = 1$, in which the model receives only $\boldsymbol{x}$. We focus on the final token. Using the same construction as in the looped TF, we can show that a single feed-forward layer followed by an attention layer suffices to copy all input tokens to every position. Specifically, the hidden representation at the last position becomes

$$h_1 = \big(\boldsymbol{e}(v_1(x)), \, 1, \, \ldots, \, \boldsymbol{e}(v_n(x)), \, 1, \, \boldsymbol{0}, \, 1, \, \boldsymbol{0}, \, \boldsymbol{e}'_0, \, \boldsymbol{0}\big), \tag{77}$$

where $v_i(x) = x_i$ for all $i \in [n]$, corresponding to the input nodes. In what follows, we consider the case $k > 1$. By the same argument as for Looped TF, based on Equation (62), we show that the feed-forward layers can encode the entire computation graph into their weights and simulate all nodes simultaneously. That is, there exist two feed-forward layers whose composition, denoted by $\mathcal{FF}$, satisfies $\mathcal{FF}(h_k) = h_{k+1}$.

Consider the last feed-forward layer applied after the zero-weight attention layer. By substituting $\boldsymbol{e}_i = \boldsymbol{e}'_k$ and $\boldsymbol{x} = (h_k)_{1:m-|\Sigma|}$ into Lemma B.6, there exists a feed-forward network FF such that

$$(\text{id} + \text{FF})(h_k) = \begin{cases} h_{k+1}, & \text{if } k < \text{depth}(G_n), \\ \big((h_{k+1})_{1:m-|\Sigma|}, \, \boldsymbol{e}(v_{o_i}(x))\big), & \text{if } k = \text{depth}(G_n) + i. \end{cases} \tag{78}$$

In particular, for all $k < \text{depth}(G_n)$, the output of the last token, for the input $(x_1, x_2, \ldots, x_n, h_1, h_2, \ldots, h_k)$ is $h_{k+1}$. For the final positions corresponding to output nodes, namely $k = \text{depth}(G_n) + i$, the hidden state $\boldsymbol{h}_k$ contains the embedding of the output symbol $v_{o_i}(x)$. Applying the output projection yields

$$\textbf{OUT}(\boldsymbol{h}_k) = \begin{bmatrix} \boldsymbol{0} & \boldsymbol{I}_{|\Sigma|} \end{bmatrix} \boldsymbol{h}_k = \boldsymbol{e}(v_{o_i}(x)). \tag{79}$$

Finally, the decoding function outputs the symbol in $\Sigma$ corresponding to the maximum coordinate of $\textbf{OUT}(\boldsymbol{h}_k)$. $\square$

## B.7. Proof for Theorem 3.12

For the upper bound $\text{Loop}[\log^k n, \text{poly}(n), 1 \text{ (resp. } \log n)] \subseteq \text{AC}^k \text{ (resp. } \text{TC}^k)$, we follow the argument of (Li et al., 2024). Their key observation is that a restricted form of automaton can model iterative computation under constant precision: the rounding operation preserves monotonicity, and constant precision yields counter-free, restricted state spaces, which are therefore computable by $\text{AC}^0$ circuits. In the case of polynomial precision, prefix summation can be simulated by $\text{TC}^0$, which also allows the detection and correction of rounding in floating-point arithmetic.

**Theorem B.9** (Li et al., 2024). *For $s(n) \in \text{poly}(n)$, $\text{sum}_{s(n)} : (\mathbb{F}_{s(n)})^n \to \mathbb{F}_{s(n)}$ is computable by $\text{TC}^0$ circuits, and by $\text{AC}^0$ circuits when $s(n)$ is constant.*

It has also been shown that gates can be efficiently simulated by feedforward layers.

**Lemma B.10** (Li et al., 2024). *Unbounded-fanin* $\text{AND}, \text{OR}$ *(resp.* $\text{MAJORITY}$*)* $: \{0,1\}^n \to \{0,1\}$ *can be simulated by a two-layer feedforward ReLU network with constant (resp.* $\log n$*) bits of precision constant hidden dimension and additional* $n$ *constant inputs of value* $1$.

*Proof for Theorem 3.12.* $\text{Loop}[\log^k n, \text{poly}(n), 1 \text{ (resp. } \log n)] \subseteq \text{AC}^k \text{ (resp. } \text{TC}^k)$ : In Transformers, constant-depth computation is defined by Summation with Iterative Rounding (see Definition B.3), which by Theorem B.9 can be simulated in $\text{AC}^0$ (resp. $\text{TC}^0$). A looped TF simply stacks these computations vertically through iteration, and thus the result follows.

$\text{AC}^k \text{ (resp. } \text{TC}^k) \subseteq \text{Loop}[\log^k n, \text{poly}(n), 1 \text{ (resp. } \log n)]$ : Since Boolean circuits are DAGs, the claim follows directly from Theorem 3.10 together with Lemma B.10. $\qquad\square$

# C. Deferred Proofs for Section 4

## C.1. Definition for Models of Computation

We model a language model as a probabilistic process that, given an input and a generated prefix, produces either an internal reasoning state or an output token. This process induces a probability distribution over final outputs. We first formalize CoT under this setting. We focus on the saturated hardmax attention as in (Merrill et al., 2022; Nowak et al., 2024).

**Definition C.1** (Language model with CoT). Let $\mathcal{V}$ be a vocabulary. Given an input $x \in \mathcal{V}^*$, a language model with CoT stochastically generates a sequence of output blocks of the form

$$\left( r_1, e, y_1, e', r_2, e, y_2, e', \cdots r_m, e, y_m, e' \right), \tag{80}$$

where each $r_i \in \mathcal{V}^*$ represents an explicit reasoning trace, $y_i \in \mathcal{V}$ is an output token, and $e, e' \in \mathcal{V}$ are special delimiter tokens. The final output is the string $y_1 \cdots y_m$. Generation proceeds autoregressively: at iteration $i$, the model generates a reasoning segment $r_i$ followed by an output token $y_i$, conditioned on the input $x$, the previously generated outputs $y_{<i}$, and prior reasoning segments $r_{<i}$. We denote by $p(y \mid x)$ the resulting distribution over final outputs.

Then we define the language models with latent thought reasoning: Coconut and looped TF.

**Definition C.2** (Language model with Coconut). Let $\mathcal{V}$ be a vocabulary. Given an input $x \in \mathcal{V}^*$, a language model with Coconut generates a sequence of blocks

$$\left( r_1, y_1, r_2, y_2, \ldots, r_m, y_m \right), \tag{81}$$

where each $r_i \in \mathbb{F}^{d^*}$ represents an internal continuous reasoning state, and each $y_i \in \mathcal{V}$ is an output token.

Generation proceeds autoregressively. At each iteration, the internal continuous reasoning state $r_i$ is generated deterministically as a function of the input $x$, the previously generated outputs $y_{<i}$, and the prior internal states $r_{<i}$, while the output token $y_i$ is generated stochastically. The decision of when to emit an output token is determined internally by the model.

For looped TF models, the computation proceeds through internally repeated iterations before producing any output tokens. The number of iterations is determined by the model as a function of the input.

**Definition C.3** (Language model with looped TF). Given an input $x \in \mathcal{V}^*$, a looped TF produces an output sequence autoregressively. At each iteration $i \in [m]$, the model internally performs a number of repeated transformation steps before emitting an output token $y_i$, conditioned on the input $x$ and the previously generated outputs $y_{<i}$. The number of internal

loop iterations required to generate each output token is determined internally by the model as a function of the input $x$ and the generated prefix $y_{<i}$.

We define complexity classes corresponding to language models under different reasoning paradigms. In contrast to the non-uniform models typically used in parallel computation analysis, we adopt a **uniform setting** analogous to Turing machines: a single model with a fixed set of parameters is applied to all input lengths, while the number of reasoning steps is allowed to grow as a function of the input size $n$. Following the convention in (Merrill & Sabharwal, 2024), we allow $O(\log n)$ bits of numerical precision to represent positional embeddings and internal activations. Furthermore, we extend the output space beyond simple binary decisions. Specifically, the model's output is not restricted to $\Sigma^k$, but can be an arbitrary finite binary string in $\Sigma^*$. This extension enables the model to represent functions of the form $f : \Sigma^* \to \mathbb{R}$, thereby capturing counting, probabilistic modeling, and approximation tasks.

**Definition C.4** (Complexity Classes pCoT, pCT, and pLOOP). Let $\mathsf{pCoT}[T(n)]$, $\mathsf{pCT}[T(n)]$, and $\mathsf{pLOOP}[T(n)]$ denote the classes of probabilistic computations that define an output distribution $p : \Sigma^* \times \Sigma^* \to [0, 1]$. A function $f$ belongs to such a class if there exists a language model $\mathcal{M}$, employing CoT, Coconut, or looped TF, respectively, such that for any input $x \in \Sigma^n$, the model induces the distribution $p(x, \cdot)$ within $O(T(n))$ steps using $O(\log n)$ bits of numerical precision.

## C.2. Lemma: Universality of Chain of Thought

**Definition C.5.** A probabilistic Turing machine $M = (Q, \Sigma, \Gamma, q_0, q_1, \delta_1, \delta_2)$ is defined as:

- $Q$ is a finite set of states,

- $\Sigma$ is the input/output alphabet,

- $\Gamma$ is the tape alphabet, with $\Sigma \subseteq \Gamma$ and a distinguished blank symbol $\sqcup \in \Gamma$,

- $q_0 \in Q$ denotes the initial state, and $q_1 \in Q$ denotes the final state,

- $\delta_1, \delta_2 : Q \times \Gamma \to Q \times \Gamma \times (\Sigma \cup \{\varepsilon\}) \times \{L, S, R\}$ are two transition functions.

At each step, $M$ chooses uniformly at random between $\delta_1$ and $\delta_2$. Each transition $\delta_i(q, a) = (q', b, \sigma, D)$ is interpreted as: writing $b \in \Gamma$ on the work tape, writing $\sigma \in \Sigma$ on the output tape, where $\varepsilon$ means that nothing is written, and moving the tape head in direction $D \in \{L, S, R\}$, where $L$ = left, $R$ = right, and $S$ = stay. The output of $M$ is defined to be the string remaining on the output tape when the machine halts.

Prior work has shown that probabilistic CoT can simulate any such PTM, thereby demonstrating its universality.

**Lemma C.6** (Nowak et al., 2024). *Let $M$ be a PTM with input and output alphabet $\Sigma$, and running time bounded by $T(n)$ on inputs of length $n$. Then there exists a log-precision CoT model with stochastic decoding, denoted $\mathsf{CoT}_M$, such that the induced output distribution of $\mathsf{CoT}_M$ coincides exactly with that of $M$. Formally, for every input string $x \in \Sigma^*$ and output string $y \in \Sigma^*$,*

$$\Pr\big[\mathsf{CoT}_M(x) = y\big] \;=\; \Pr\big[M(x) = y\big]. \tag{82}$$

*Moreover, the number of reasoning steps of $\mathsf{CoT}_M$ is bounded by $\mathsf{poly}(|x|)$.*

## C.3. Self-Reducibility and Complexity of Approximate Counting

Here, we provide the definitions of relation and associated counting problems.

**Definition C.7** (Relation). A *relation* over an alphabet $\Sigma$ is a subset $R \subseteq \Sigma^* \times \Sigma^*$. For an input $x \in \Sigma^*$, we denote

$$R(x) := \{\, y \in \Sigma^* : (x, y) \in R \,\}.$$

**Definition C.8** (Counting). Given a relation $R$, the associated counting function is defined as

$$N_R : \Sigma^* \to \mathbb{N}, \qquad N_R(x) := |R(x)|.$$

**Definition C.9** (*p*-relation). A relation $R \subseteq \Sigma^* \times \Sigma^*$ is called a *p-relation* if

- the membership $(x, y) \in R$ can be decided in time polynomial in $|x|$, and
- there exists a polynomial $p$ such that for all $(x, y) \in R$, we have $|y| \leq p(|x|)$.

**Definition C.10.** The *extension counting function* associated with a relation $R \subseteq \Sigma^* \times \Sigma^*$ is

$$\text{EXT}_R : \Sigma^* \times \Sigma^* \to \mathbb{N}, \qquad \text{EXT}_R(x, w) := \big|\{\, z \in \Sigma^* : (x, wz) \in R \,\}\big|.$$

The complexity class $\#\text{P}$ consists of counting problems associated with $p$-relations (Valiant, 1979). Then, we provide definitions of schemes for approximate counting. In the remainder of this paper, we say that an algorithm produces an output *approximating $f(x)$ within ratio $1 + \varepsilon$* if its output $\hat{f}(x)$ satisfies

$$(1 - \varepsilon)f(x) \leq \hat{f}(x) \leq (1 + \varepsilon)f(x),$$

**Definition C.11** (FPTAS). An algorithm is called a *fully polynomial-time approximation scheme* (FPTAS) for a function $f$ if, for any input $x$ and any $\varepsilon > 0$, it produces an output approximating $f(x)$ within ratio $1 + \varepsilon$, and runs in time polynomial in $|x|$ and $1/\varepsilon$.

**Definition C.12** (FPRAS). An algorithm is a *fully polynomial-time randomized approximation scheme* (FPRAS) for a function $f$ if, for any input $x$ and any $\varepsilon > 0$ and $\delta > 0$, it produces an output approximating $f(x)$ within ratio $1 + \varepsilon$ with probability at least $1 - \delta$, and runs in time polynomial in $|x|$, $1/\varepsilon$, and $\log(1/\delta)$.

The following proposition formalizes the relationship between approximate counting and the extension counting function.

**Proposition C.13** (Jerrum et al., 1986). *Let $R$ be a $p$-relation and let $x \in \Sigma^n$. Let $m = p(n)$ and define $r = 1 + \frac{\varepsilon}{2m}$. If there exists a FPRAS that approximates $\text{Ext}_R(x, w)$ within ratio $r$ for all prefixes $w$ with probability at least $1 - \delta/m$, then there exists an FPRAS that approximates $N_R(x)$ within ratio $1 + \varepsilon$ with probability at least $1 - \delta$.*

This reduction transforms the global counting problem into a sequence of local estimation steps.

**Proposition C.14.** *For any $0 < \varepsilon \leq 1$ and any integer $m \geq 1$, it holds that*

$$\left(1 + \frac{\varepsilon}{2m}\right)^m < 1 + \varepsilon. \tag{83}$$

*Proof.* We use the standard inequality $1 + x \leq e^x$, which holds for all $x \in \mathbb{R}$. Substituting $x = \frac{\varepsilon}{2m}$, we have:

$$\left(1 + \frac{\varepsilon}{2m}\right)^m \leq \left(\exp\left(\frac{\varepsilon}{2m}\right)\right)^m = e^{\varepsilon/2}. \tag{84}$$

For $0 < \varepsilon \leq 1$, we show that $e^{\varepsilon/2} < 1 + \varepsilon$. Let $f(\varepsilon) = 1 + \varepsilon - e^{\varepsilon/2}$. Then $f(0) = 0$ and $f'(\varepsilon) = 1 - \frac{1}{2}e^{\varepsilon/2}$. Since $e^{\varepsilon/2} \leq e^{1/2} < 2$ for all $\varepsilon \in [0, 1]$, it follows that $f'(\varepsilon) > 0$. Thus, $f$ is strictly increasing on $[0, 1]$, implying $f(\varepsilon) > f(0) = 0$, or equivalently $e^{\varepsilon/2} < 1 + \varepsilon$. $\square$

*Proof for Proposition C.13.* Let $x \in \Sigma^n$ and $m = p(n)$. We express $N_R(x)$ as a product of $m$ ratios. Let $w = y_1 y_2 \ldots y_m$ be a witness, and let $w^{(i)}$ denote its prefix of length $i$. We can write:

$$N_R(x) = \text{Ext}_R(x, \lambda) = \prod_{i=1}^{m} \frac{\text{Ext}_R(x, w^{(i-1)})}{\text{Ext}_R(x, w^{(i)})} \cdot \text{Ext}_R(x, w^{(m)}). \tag{85}$$

In the standard self-reducibility framework, we estimate each ratio $\rho_i = \text{Ext}_R(x, w^{(i-1)})/\text{Ext}_R(x, w^{(i)})$ using the assumed approximation scheme. Let $A_i$ be the estimator for the $i$-th ratio such that:

$$\mathbb{P}\left[\frac{1}{r} \leq \frac{A_i}{\rho_i} \leq r\right] \geq 1 - \frac{\delta}{m}. \tag{86}$$

By the union bound, the probability that at least one of these $m$ estimations fails to stay within the ratio $r$ is at most $m \cdot (\delta/m) = \delta$. Therefore, with probability at least $1 - \delta$, all $m$ estimations are successful. Under this condition, the final estimate $\hat{N} = \prod_{i=1}^{m} A_i$ satisfies:

$$\frac{1}{r^m} \leq \frac{\hat{N}}{N_R(x)} \leq r^m. \tag{87}$$

From Proposition C.14, we have $r^m = (1 + \frac{\varepsilon}{2m})^m < 1 + \varepsilon$. Furthermore, for $0 < \varepsilon \leq 1$, it holds that $1/r^m > (1+\varepsilon)^{-1} \geq 1 - \varepsilon$, ensuring a relative error of at most $\varepsilon$.

Regarding complexity, each of the $m$ calls to the FPRAS for $\text{Ext}_R$ runs in time $\text{poly}(n, (r-1)^{-1}, \log(m/\delta))$. Substituting $r - 1 = \varepsilon/2m$, the runtime per call is $\text{poly}(n, m/\varepsilon, \log(m/\delta))$. Since $m = p(n)$ is a polynomial in $n$, the total running time is $\text{poly}(n, 1/\varepsilon, \log(1/\delta))$, which satisfies the requirements for an FPRAS. □

We have shown that approximate counting reduces to approximating the *extension counting function*. For a general relation $R$, however, the connection between the extension counting function and the original counting problem for $R$ remains unclear. This gap is bridged by the notion of *self-reducibility*: for self-reducible relations, the extension counting function can be reduced back to the original counting problem for $R$.

**Definition C.15** (Schnorr, 1976). A relation $R \subseteq \Sigma^* \times \Sigma^*$ is *self-reducible* if:

1. There exists a polynomial-time computable function $g \in \Sigma^* \to \mathbb{N}$ s.t., $(x, y) \in R \Rightarrow |y| = g(x)$;

2. There exists a polynomial-time Turing machine that decides membership in $R$.

3. There exist polynomial-time computable functions $\psi \in \Sigma^* \times \Sigma^* \to \Sigma^*$ and $\sigma \in \Sigma^* \to \mathbb{N}$ s.t.

$$\sigma(x) = O(\log|x|), \tag{88}$$
$$g(x) > 0 \Rightarrow \sigma(x) > 0 \quad \forall x \in \Sigma^*, \tag{89}$$
$$|\psi(x, w)| \leq |x| \quad \forall x, w \in \Sigma^*, \tag{90}$$

and such that, for all $x \in \Sigma^*$, $y = y_1 \dots y_n \in \Sigma^*$,

$$\langle x, y_1, \dots, y_n \rangle \in R \iff \langle \psi(x, y_1 \dots y_{\sigma(x)}), y_{\sigma(x)+1}, \dots, y_n \rangle \in R. \tag{91}$$

For example, SAT is self-reducible: by fixing a prefix of variables and applying the reduction map $\psi$, the problem is simplified to a smaller instance whose solutions extend the chosen prefix. Consider the Boolean formula $F = (x_1 \lor x_2) \land (\neg x_1 \lor x_3) \land (\neg x_2 \lor \neg x_3)$, and suppose we fix the first variable to $x_1 = 1$. The residual instance is obtained by applying $\psi(F, (1))$, which substitutes $x_1 = 1$ and simplifies the formula by deleting satisfied clauses and removing falsified literals: $\psi(F, (1)) = (x_3) \land (\neg x_2 \lor \neg x_3)$. The unit clause $(x_3)$ forces $x_3 = 1$, which in turn simplifies $(\neg x_2 \lor \neg x_3)$ to $\neg x_2$, yielding $x_2 = 0$. Hence the unique residual assignment is $(x_2, x_3) = (0, 1)$, and together with the prefix $x_1 = 1$, we obtain the satisfying assignment $(x_1, x_2, x_3) = (1, 0, 1)$.

For self-reducible relations, the extension counting function is no harder to approximate than the original counting problem.

**Proposition C.16** (Jerrum et al., 1986). *Let $R$ be self-reducible. If there exists an FPRAS for $N_R$, then there exists an FPRAS for $\text{Ext}_R$.*

*Proof.* By the definition of self-reducibility, the extension function $\text{Ext}_R(x, w)$, which counts the number of strings $y$ such that $(x, wy) \in R$, can be mapped to the counting problem of a modified instance. Specifically, for any prefix $w$ where $|w| \leq \sigma(x)$, there exists a polynomial-time computable mapping $\psi$ such that:

$$\text{Ext}_R(x, w) = |\{y \in \Sigma^{\sigma(x) - |w|} : (x, wy) \in R\}| = N_R(\psi(x, w)). \tag{92}$$

Since $R$ is self-reducible, the instance $x_w = \psi(x, w)$ can be constructed in polynomial time relative to $|x|$. By the hypothesis, there exists an FPRAS for $N_R$, which provides a randomized $(1 \pm \epsilon)$-approximation of $N_R(x_w)$ in time polynomial in $|x_w|$ and $1/\epsilon$. Consequently, this algorithm serves as an FPRAS for $\text{Ext}_R(x, w)$, as it runs in polynomial time and satisfies the required approximation guarantees. □

## C.4. Proof for Theorem 4.3

**Lemma C.17** (Formal Statement of Lemma 4.3). *Assume that* FPTAS $\subsetneq$ FPRAS *for self-reducible relations. There exists a self-reducible relation $R$ and an associated function $\text{Ext}_R : \Sigma^* \times \Sigma^* \to \mathbb{N}$ defined by $\text{Ext}_R(x, y_{<i}) := |\{ z \in \Sigma^* : (x, y_{<i}z) \in R \}|$ such that language models with CoT using polynomially many reasoning steps, which output a distribution for a given input $(x, y_{<i}) \in \Sigma^n \times \Sigma^*$ by using a linear head for the last hidden state before emitting the output token, admit an FPRAS for $\text{Ext}_R$, whereas no latent thought with polynomially many iterations admits the same approximation guarantee using a linear head for the last hidden state before emitting the output token.*

*Proof.* By Lemma C.6, CoT can simulate any probabilistic Turing machine running in polynomial time; thus, it can implement an FPRAS for $N_R$. By Proposition C.16, the existence of an FPRAS for $N_R$ further implies the existence of an FPRAS for the extension function $\mathrm{Ext}_R$. On the other hand, latent thought consisting of a polynomial number of iterations can always be simulated by a deterministic polynomial-time Turing machine, provided that all state transitions in the latent computation are deterministic. Consequently, if such a latent thought process were to admit an FPRAS for $\mathrm{Ext}_R$, it would effectively yield a deterministic polynomial-time approximation scheme. By Proposition C.16, the existence of such a scheme for $\mathrm{Ext}_R$ would imply the existence of an FPTAS for $N_R$. However, under the standard complexity-theoretic assumption that $\mathsf{FPTAS} \subsetneq \mathsf{FPRAS}$ for self-reducible relations, there exists a self-reducible relation $R$ whose counting function admits an FPRAS but no FPTAS. This yields a contradiction. $\square$

### C.5. Proof for Theorem 4.4

**Definition C.18** (FPAUS). Uniform generation asks to sample an element $y$ uniformly at random from $R(x)$. A *fully polynomial almost uniform sampler* (FPAUS) for $R$ is a randomized algorithm that, given an input $x \in \Sigma^*$ and an accuracy parameter $\varepsilon > 0$, runs in time polynomial in $|x|$ and $\log(1/\varepsilon)$, and outputs a distribution $q(\cdot \mid x)$ such that

$$\big\| q(\cdot \mid x) - U(R(x)) \big\|_{\mathrm{TV}} \leq \varepsilon,$$

where $U(R(x))$ denotes the uniform distribution over the set $R(x)$, and $\| \cdot \|_{\mathrm{TV}}$ denotes total variation distance.

For self-reducible relations, the following holds.

**Theorem C.19** (Jerrum et al., 1986). *Let $R$ be a self-reducible relation. There exists an FPRAS for approximating $|R(x)|$ if and only if there exists an FPAUS for sampling uniformly from $R(x)$.*

*Proof of Theorem 4.4.* The target uniform conditional distribution is defined as follows:

$$p(y_i \mid x, y_{<i}) := \frac{\mathrm{EXT}_R(x, y_{<i} y_i)}{\sum_{u \in \Sigma} \mathrm{EXT}_R(x, y_{<i} u)} \qquad (y_i \in \Sigma). \tag{93}$$

Assume an FPRAS $\mathcal{A}(x, \varepsilon, \delta)$ exists for the self-reducible relation $|R(x)|$. By Proposition C.16, the existence of an FPRAS for $|R(x)|$ implies the existence of an FPRAS for the extension function $\mathrm{EXT}_R$. We construct a CoT that samples $y \in R(x)$ by sequentially approximating these conditional probabilities. For each step $i \in \{1, \ldots, m(n)\}$, the CoT computes $(1 \pm \varepsilon)$-accurate estimates $\widehat{\mathrm{EXT}}_R(x, y_{<i} u)$ for all $u \in \Sigma$ using $\mathcal{A}$, and induces the following distribution:

$$\pi(y_i \mid x, y_{<i}) := \frac{\widehat{\mathrm{EXT}}_R(x, y_{<i} y_i)}{\sum_{u \in \Sigma} \widehat{\mathrm{EXT}}_R(x, y_{<i} u)}. \tag{94}$$

Conditioned on the event that all estimates in Equation (94) are $(1 \pm \varepsilon)$-accurate, the multiplicative error is bounded by:

$$\frac{1 - \varepsilon}{1 + \varepsilon} \leq \frac{\pi(y_i \mid x, y_{<i})}{p(y_i \mid x, y_{<i})} \leq \frac{1 + \varepsilon}{1 - \varepsilon}. \tag{95}$$

To ensure the cumulative approximation error remains within $(1 \pm \varepsilon')$, we set the local precision to $\varepsilon \leq \frac{\varepsilon'}{2 + \varepsilon'}$, which yields $\frac{1+\varepsilon}{1-\varepsilon} \leq 1 + \varepsilon'$ and $\frac{1-\varepsilon}{1+\varepsilon} \geq 1 - \varepsilon'$. To ensure the failure probability is at most $\delta'$, we apply a union bound over the $m(n)$ generation steps and the $|\Sigma|$ calls per step. By setting the local confidence to $\delta \leq \frac{\delta'}{m(n)(|\Sigma|+1)}$, the joint success event holds with probability at least $1 - \delta'$. Under this event, the CoT correctly simulates an FPRAS for $p(y_i \mid x, y_{<i})$ in total time $\mathrm{poly}(n, 1/\varepsilon', \log(1/\delta'))$. Finally, since CoT can represent an FPRAS by Lemma 4.3, it satisfies the requirements for the construction.

On the other hand, we show that latent thought cannot compute such an approximation. Suppose, for contradiction, that the model could compute the conditional distribution $\pi(y_i \mid x, y_{<i})$ to within a $(1 \pm \varepsilon)$ relative error in a single step. We define the estimator for the total count $Z(x) = |R(x)|$ as:

$$\widehat{Z}(x) := \left( \prod_{i=1}^{m(n)} \pi(y_i \mid x, y_{<i}) \right)^{-1}. \tag{96}$$

Since the true distribution satisfies $p(y \mid x) = 1/|R(x)| = \prod_i p(y_i \mid x, y_{<i})$, the relative error of $\widehat{Z}(x)$ is governed by the product of local errors:

$$(1+\varepsilon)^{-m(n)}|R(x)| \leq Z(x) \leq (1-\varepsilon)^{-m(n)}|R(x)|, \tag{97}$$

By setting $\varepsilon \leq \frac{\varepsilon'}{2m(n)}$, we apply Proposition C.14 and the properties of multiplicative error:

$$(1+\varepsilon)^{-m(n)} \geq 1 - m(n)\varepsilon \geq 1 - \varepsilon'/2 > 1 - \varepsilon', \tag{98}$$

and for sufficiently small $\varepsilon$,

$$(1-\varepsilon)^{-m(n)} \leq 1 + 2m(n)\varepsilon \leq 1 + \varepsilon'. \tag{99}$$

Substituting these into Equation (96), we obtain:

$$(1-\varepsilon')|R(x)| \leq \widehat{Z}(x) \leq (1+\varepsilon')|R(x)|. \tag{100}$$

This implies that if the model could compute $\pi$ accurately, $\widehat{Z}(x)$ would constitute an FPTAS for $|R(x)|$. However, under the standard complexity-theoretic assumption that FPTAS $\subsetneq$ FPRAS for self-reducible relations, this yields a contradiction. $\square$

## D. Experimental Details

### D.1. Fundamental Algorithmic Reasoning Tasks

#### D.1.1. TASK SETTINGS

**Word Problem**  We define a sequence prediction task based on finite groups such as the symmetric group $S_5$. Given a sequence of group elements of length $k$, the model is required to output the cumulative products obtained by scanning the sequence from left to right. Formally, for an input sequence $(g_1, g_2, \ldots, g_k)$, the target sequence is $(g_1,\ g_1g_2,\ g_1g_2g_3,\ \ldots,\ g_1g_2\cdots g_k)$. We follow the setting of (Merrill & Sabharwal, 2025b).

**Connectivity**  To ensure that the reachability labels are approximately balanced, we generate undirected graphs according to the Erdős–Rényi model (Erdos & Renyi, 1959) $G(n, p)$, where $n$ is the number of vertices and each possible edge is included independently with probability $p$. In the supercritical regime ($pn = c > 1$), a single "giant" connected component emerges, occupying a fraction $s \in (0, 1)$ of the vertices, which satisfies $s = 1 - e^{-cs}$. Consequently, the probability that two uniformly random vertices are both in this component—and hence mutually reachable—is approximately $s^2$. To target a reachability probability of $1/2$, we set $s \approx \sqrt{\frac{1}{2}} \approx 0.707, c \approx \frac{-\ln(1-s)}{s} \approx 1.74$, and thus $p = \frac{c}{n} \approx \frac{1.7}{n}$. In practice, for each graph of size $n$ we fix $p = 1.7/n$, which empirically yields $\Pr[\text{reachable}] \approx 50\%$ for $n \in [50, 100]$. We follow the encoding scheme of Sanford et al. (2024a). The input to the model is serialized as a flat token sequence consisting of three parts: $v_0\, v_1\, \cdots\, v_{n-1}\, e_1\, e_2\, \cdots\, e_m\, s, t$ where each vertex is denoted by a token $v_i$, each edge is represented as a pair "`u, v`" with $u < v$, and the final token "`s, t`" specifies the source–target pair for the reachability query.

**Arithmetic Expression Evaluation**  Following (Feng et al., 2023), we generate expressions over integers modulo $r$ using the four operations $+, -, \times, \div$, where multiplication and division are defined via precomputed modular tables. To guarantee that each expression evaluates to a specific target value, we grow expressions *backwards*: starting from a sampled number, we iteratively replace it with a binary sub-expression that preserves its value under modular arithmetic. Different from (Feng et al., 2023), we fix the modulus to $r = 3$, as our focus lies in evaluating the reasoning over expressions rather than exploring the properties of each modular arithmetic system.

**Edit Distance**  The Edit Distance task requires computing the minimum number of edit operations needed to transform one string into another. The allowed operations are insertion, deletion, and replacement of a single character, and the objective is to predict the total edit distance given two input strings. To build the dataset, we follow (Feng et al., 2023). We first generate two strings over a randomly sampled alphabet. The first string has a fixed length, while the second string is produced in two possible ways: with probability $0.4$, it is drawn as a random string of nearly the same length (within $\pm 3$ characters), and with probability $0.6$, it is derived from the first string by applying a given number of random edit operations. Each edit operation is chosen uniformly among deletion, replacement, and insertion. To avoid trivial cases, string pairs that are identical or whose lengths differ excessively are rejected and resampled. Finally, the shorter string is always placed first to maintain a consistent input format. An example instance in the format is shown below:

*Table 3.* Results for CoT on parallelizable tasks trained with uniform selection.

| Task | $n$ | Chain of Thought | | | |
| --- | --- | --- | --- | --- | --- |
| | | 8 | 16 | 32 | 64 |
| Word Problem | 64 | 0.8 | 0.8 | 0.8 | **100.0** |
| Graph Connectivity | 32 | 81.0 | 81.4 | 83.6 | **100.0** |
| Arithmetic Evaluation | 16 | 41.0 | 41.5 | 41.2 | **82.5** |
| Edit Distance | 16 | 69.2 | 70.3 | 82.6 | **94.8** |

`s v d h s s e e ... v e | s h d s s s s ... e s e <sep> 20` Here, the two input strings are separated by the token "`|`", "`<sep>`" marks the end of the inputs, and the final number "`20`" denotes the computed edit distance.

### D.1.2. TRAINING CONFIGURATION

**Configuration of chain of thought**  For CoT models, training is performed with supervision of step-by-step algorithms. **(1) Word problem:** for this task, the CoT algorithm proceeds by sequentially scanning the token sequence and producing at each prefix the evaluation result of the expression step by step. Thus, the overall length of the CoT sequence matches the input length. **(2) Graph connectivity:** Following Bavandpour et al. (2025), the algorithm sequence is simply the trace of a breadth-first search (BFS) starting from the source $s$. At each step, the model emits the incident edges of the currently expanded node in the order they are visited. The sequence terminates as soon as the target $t$ is discovered. To implement this algorithm, we maintain a list ("scratchpad") initialized with a dummy marker and the source, $(\mathtt{N}, s)$. We iterate through this list from left to right (i.e., queue order). Whenever the current node $u$ is expanded, we append to the end of the list all incident edges $(u, v)$ for neighbors $v$, followed by a separator token $(u, \mathtt{N})$. **(3) Arithmetic expression evaluation:** Following (Feng et al., 2023), the CoT takes the fully expanded expression and repeatedly evaluates one innermost subexpression, writing down the simplified expression at each step until only a single numeral remains. For example, $2 * (0 + 1)/2 \to 2 * 1/2 \to 2/2 \to 1$. The overall CoT sequence has quadratic length. **(4) Edit distance:** Following (Feng et al., 2023), the CoT algorithm outputs the DP table entries in the same order they are computed, i.e., row by row from top-left to bottom-right (topological order). This yields a quadratic number of steps in the input length.

**Optimization and model details.**  We trained all models using the AdamW optimizer with a linear learning rate schedule. The initial learning rate was set to $1 \times 10^{-4}$ with a weight decay of $0.01$, and a batch size of $256$. Training was continued until the training loss plateaued. For looped TFs, curriculum learning was applied to all tasks except edit distance: the input size was increased by $2$ for the word problem task, and by $4$ for the connectivity and arithmetic evaluation tasks. The model architecture was based on standard Transformers with an embedding dimension of $256$. We used $4$ attention heads, and varied the number of Transformer layers depending on the task: two layers for word problems, a single layer for connectivity, and time-modulated (Xu & Sato, 2025) model with a single layer for looped TF, to stabilize training, on both arithmetic evaluation and the edit distance task. For CoT, we use the same configuration of the Transformer block.

**Uniform selection.**  To estimate a lower bound on the number of reasoning steps required by CoT for each task, we first follow the procedure introduced in prior work (Bavandpour et al., 2025). Specifically, given a complete CoT trajectory consisting of $T$ intermediate reasoning steps, we construct shortened trajectories by uniformly selecting $k$ step indices from $\{1, \ldots, T\}$, i.e., by taking a uniformly spaced subsequence of the original trajectory. Only the selected intermediate steps are used, while the remaining steps are removed. We then evaluate task performance as a function of $k$, as shown in Table 3.

**Stepwise internalization.**  We adopt stepwise internalization proposed by (Deng et al., 2024), a curriculum-based training procedure that gradually removes CoT tokens and encourages the model to internalize intermediate reasoning within its hidden states. Starting from a model trained on full CoT trajectories, we progressively truncate intermediate reasoning tokens according to a predefined schedule and finetune the model at each stage. Specifically, when the CoT length is greater than $128$, we remove $16$ tokens per stage until the remaining CoT length reaches $128$. We then continue removing $8$ tokens per stage until the CoT length is reduced to $8$, training the model for $16$ epochs at each stage. The results for each fundamental algorithmic reasoning task are shown in Fig. 10.

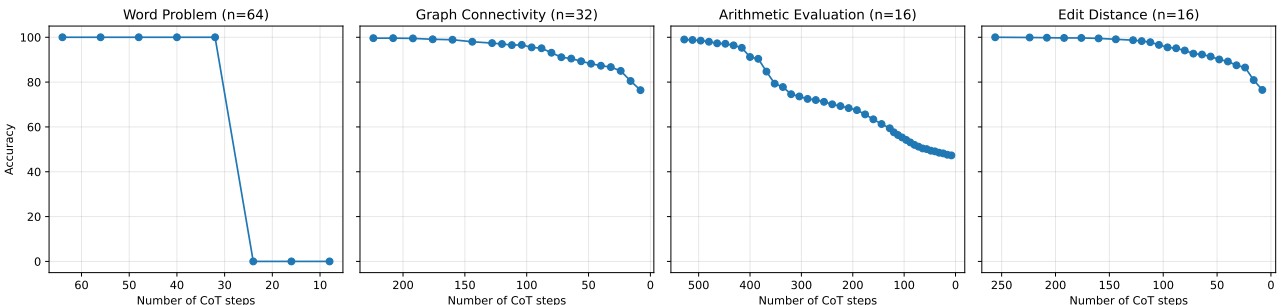

*Figure 10.* Accuracy as a function of the CoT steps during stepwise internalization.

## D.2. Approximate Counting and Approximate Sampling

### D.2.1. APPROXIMATE COUNTING OF DNF FORMULAS

To generate the dataset, we first construct a DNF formula $F$ by sampling $m$ clauses, each consisting of $w$ distinct literals over $n$ Boolean variables. Each literal is independently assigned to be either positive or negated. The formula is then serialized into a token sequence in which each clause is represented by its index together with variable–value pairs such as "2 = +1" or "4 = −1". For looped TFs, we prepare 100,000 training samples and 1,000 test samples. For CoT models, we instead generate an online dataset with the following structure. To train the CoT models, we simulate a single trial of the randomized counting algorithm of Karp & Luby (1983). The sequence concatenates the serialized formula, the sampled clause, the full assignment, and the verification outcome, separated by `<sep>` tokens and terminated by `<eos>`. CoT model is trained in an autoregressive manner, where prediction targets are defined by shifting the token sequence while masking out the formula description. We trained the models using the AdamW optimizer with a linear learning rate schedule. The initial learning rate was set to $1 \times 10^{-4}$, with a weight decay of 0.01. We used a batch size of 256 (reduced to 32 for the 1000-loop setting) and trained for 10,000 iterations. For inference, we count the total number of iterations used for summing output tokens (steps) in CoT across trials, and the number of loop iterations in the looped TF.

### D.2.2. APPROXIMATE SAMPLING OF GRAPH COLORINGS

We consider the problem of *approximately sampling* a proper $k$-coloring of a graph, also known as *almost-uniform generation*, a canonical randomized task closely related to #P-hard counting problems. Given an undirected graph $G = (V, E)$ with $n = |V|$ vertices and maximum degree $\Delta$, a *proper $k$-coloring* is an assignment of colors from $\{1, \ldots, k\}$ to vertices such that no adjacent vertices share the same color. Let $\Omega_k(G)$ denote the set of all proper $k$-colorings of $G$. We restrict attention to graphs of bounded degree and assume $k \geq 2\Delta + 1$. Under this condition, classical results in approximate counting show that the number of proper $k$-colorings admits a FPAUS (Jerrum, 1995). The approximation relies on Markov Chain Monte Carlo (MCMC) sampling using Glauber dynamics for graph colorings. Starting from an arbitrary proper coloring, the Markov chain repeatedly selects a vertex uniformly at random and proposes to recolor it with a randomly chosen color, accepting the update only if the resulting coloring remains proper. When $k \geq 2\Delta + 1$, this Markov chain is known to be rapidly mixing, converging to the uniform distribution over $\Omega_k(G)$ in polynomial time. For our experiments, we generate an undirected Erdős–Rényi random graph, where each edge is included independently with probability $p = \frac{1.7}{n}$, using a fixed random seed for reproducibility. For simplicity and ease of analysis, we focus on small graphs with $n = 3$ and set the number of colors to $k = 5$.

Each sample is generated by running $T$ steps of Glauber dynamics on the space $\Omega_k(G)$ of proper $k$-colorings. Starting from a greedy proper initialization, at each step we uniformly select a vertex and a color; the recoloring is accepted if and only if it preserves properness. The final state of the Markov chain is treated as an approximate sample from the uniform distribution over $\Omega_k(G)$. In addition to the final coloring, we optionally record the entire sequence of proposals and accept/reject outcomes for CoT and use it as supervision, which is not included for latent thought. To train sequence models, we serialize the graph structure, the initial coloring, and either the MCMC history or the final coloring into a single token sequence. We evaluate the distribution of solutions generated by the model rather than only solution accuracy. Given an input $x$, we draw $N$ independent samples from the model using ancestral decoding. Generation continues until an end-of-sequence (EOS) token is produced, and from each generated sequence we extract the final $n$ tokens immediately

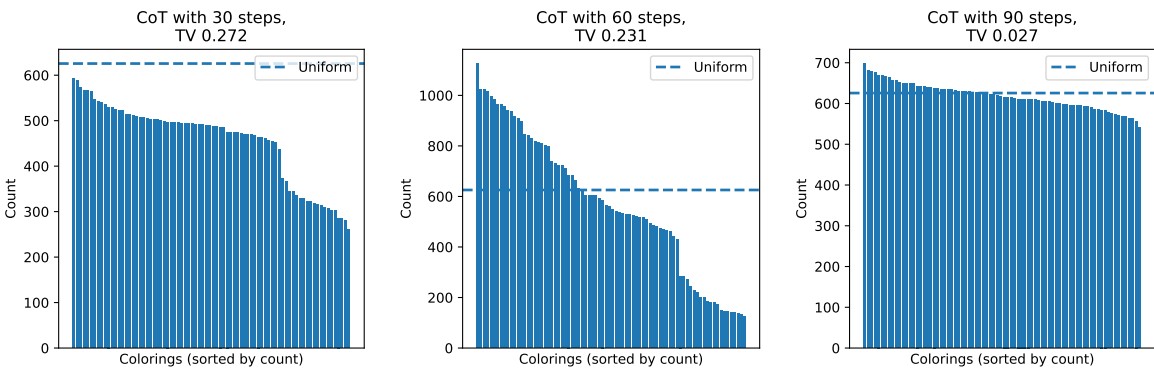

*Figure 11.* Output distributions of CoT compared against the uniform target distribution.

preceding the first EOS token, which encode a candidate solution. The resulting samples define an empirical distribution $\hat{P}$ over generated solutions via normalized occurrence counts. For each instance, the task provides an exact enumeration $\mathcal{Y}$ of all valid solutions. We define the reference distribution $P_{\text{true}}$ as the uniform distribution over this solution set, assigning probability $1/|\mathcal{Y}|$ to each $y \in \mathcal{Y}$ and zero otherwise. We quantify the discrepancy between the empirical distribution $\hat{P}$ and the uniform reference distribution using the total variation distance $\frac{1}{2} \sum_{y \in \mathcal{Y} \cup \hat{y}} \left| \hat{P}(y) - P_{\text{true}}(y) \right|$. We trained the models using the AdamW optimizer with a linear learning rate schedule. The initial learning rate was set to $1 \times 10^{-4}$, with a weight decay of $0.01$. We used a batch size of $256$ and trained for $5{,}000$ iterations. For inference, we measure the average number of steps (loops) per generation. We generate $N = 50{,}000$ samples for CoT and $N = 10{,}000$ samples for looped TFs. The resulting histograms of CoT outputs over the target support are shown in Figure 11.

