# OpenReview forum: "A Formal Comparison Between Chain of Thought and Latent Thought"
_ICML.cc/2026/Conference — ICML 2026 regular_

### Official Review · Reviewer_y1Ks · 2026-03-07

**Soundness:** 3
**Presentation:** 3
**Significance:** 3
**Originality:** 3
**Overall Recommendation:** 4
**Confidence:** 2

**Summary:**

This paper studies the complexity of two reasoning paradigms, Chain-of-Thought (CoT) and latent reasoning. The main findings are twofold: first, latent reasoning is theoretically more efficient for parallel computation due to its ability to perform iterative computation in latent space, while CoT remains inherently sequential; second, CoT enjoys an advantage on approximate counting because stochastic decoding allows sampling-based estimation. The paper further designs reasoning tasks for both settings to empirically examine these theoretical distinctions.

**Compliance With Llm Reviewing Policy:**

Affirmed.

**Final Justification:**

I maintain my original recommendation of weak accept. The paper
  provides solid and clearly presented theoretical insights
  comparing CoT and latent reasoning, with good soundness,
  significance, and originality. My main concerns were about the
  practical interpretation of latent-reasoning parallelism, the
  connection to realistic reasoning tasks, and some presentation/
  experiment clarifications. The rebuttal addressed these points
  well by clarifying that the analysis characterizes potential
  representational capability rather than guaranteed behavior of
  trained models, explaining when latent reasoning versus CoT may
  be preferable, and improving the interpretation of the
  approximate-counting experiments. Overall, the rebuttal fully
  resolved my concerns, but I keep the weak accept score given
  the still limited scope of the theoretical problem settings.

**Key Questions For Authors:**

1. It seems that the stochastic setting dominates in practice, does the result in this paper suggest that CoT can perform better than latent CoT under this setting?
2. Beyond the specific problem formulations studied in this paper, it would be helpful if the authors could discuss what characteristics of real-world reasoning tasks correspond to the two categories considered here, and how the theoretical results may inform model choice in practice. In particular, what task attributes make a problem closer to the “fundamental algorithmic reasoning” setting versus the “approximate counting” setting used in the experiments?

**Limitations:**

yes

**Strengths And Weaknesses:**

## Strengths
1. This paper presents solid theoretical results from two perspectives to explain the separation of CoT and latent reasoning, which provides novel insights in understanding the difference between the two reasoning paradigms to the community.
2. The presentation of the paper is well organized, all illustrations are clear and easy to understand at first sight, and the authors provide proof sketch for most of the theorems in the main context, which assists reading and comprehending.

## Weaknesses
1. A major concern is the formulation of latent reasoning in this paper. The analysis appears to attribute to latent reasoning a form of parallelism at the level of reasoning nodes, whereas in practice latent reasoning may only provide parallelism at the token or hidden-state level, which does not necessarily correspond to parallel reasoning paths.
2. I think the authors could further discuss how the theoretical results may guide practical model design. Since the problem classes studied in this paper are still somewhat limited, it would be helpful to clarify how these insights might extend to more realistic reasoning tasks.
3. There are some unclear points. Should m=4 instead of 3 in Figure 2(a)? From my perspective, I think the authors could place all subcaptions directly under the corresponding subfigures in Figure 2, as the current layout makes the boundaries between the three subfigures somewhat unclear. In the experimental section, I also find the results on the approximate counting tasks less illustrative than those on the algorithmic tasks. For example, what does the TV distance metric represent in this setting, and what exactly does the probability mass graph suggest? Does it indicate that CoT covers a more diverse set of valid colorings?

---

> ### Author Rebuttal · Authors · 2026-03-28
>
> > W1. A major concern is the formulation of latent reasoning in this paper. The analysis appears to attribute to latent reasoning a form of parallelism at the level of reasoning nodes, whereas in practice latent reasoning may only provide parallelism at the token or hidden-state level, which does not necessarily correspond to parallel reasoning paths.
>
> A. We thank the reviewer for this insightful comment. We would like to clarify that **parallelism at the token level, this low-level parallelism enables the representation of parallel reasoning paths in latent states**. Our formulation focuses on the representational capacity of latent reasoning. In particular, we argue that token-level parallelism provides a mechanism to represent and process DAG nodes simultaneously within latent states, which is not available in CoT.
>
> However, this does not imply that trained models necessarily learn or utilize such parallel reasoning structures in practice. Whether training dynamics lead to their emergence remains an open question. Our analysis should therefore be interpreted as characterizing the potential capabilities of latent reasoning, rather than asserting that these capabilities are fully realized in current models.
>
> > Q2. Beyond the specific problem formulations studied in this paper, it would be helpful if the authors could discuss what characteristics of real-world reasoning tasks correspond to the two categories considered here, and how the theoretical results may inform model choice in practice. In particular, what task attributes make a problem closer to the “fundamental algorithmic reasoning” setting versus the “approximate counting”?
> > W2. I think the authors could further discuss how the theoretical results may guide practical model design. Since the problem classes studied in this paper are still somewhat limited, it would be helpful to clarify how these insights might extend to more realistic reasoning tasks.
>
> A. Our problem settings can be related to real-world problems as follows:
> - $TC^d$ and fundamental algorithmic reasoning: **“Easy” problems**, which admit efficient and structured solutions (e.g., basic arithmetic such as multiplication)
> - Approximate counting: **“Hard” problems**, where obtaining the correct answer in a single deterministic pass is difficult (e.g., mathematical olympiad problems).
>
> Our results suggest that latent reasoning is well-suited for the former, where computation can be parallelized or efficiently structured, whereas CoT reasoning is more effective for the latter, where randomized computation, enabling exploration over multiple reasoning paths, is beneficial.
>
> As pointed out by the reviewer, we did not clearly present the practical implications. We therefore revise the statement in the conclusion to better clarify how our theoretical results inform practical model design.
>
> **Before:** These insights offer practical guidance for reasoning model design.
> **After:** Our results provide practical guidance for selecting between reasoning paradigms: latent reasoning is more suitable for problems that can be solved efficiently, whereas CoT is more effective for more complex problems.
>
> > W3-1. There are some unclear points. From my perspective, I think the authors could place all subcaptions directly under the corresponding subfigures in Figure 2.
>
> A. We will correct the value of $m$ in Figure 2(a) and revise the layout by placing each subcaption directly under the corresponding subfigure. We appreciate the reviewer’s suggestion to improve the clarity and readability of the paper.
>
> > W3-2. In the experimental section, I also find the results on the approximate counting tasks less illustrative than those on the algorithmic tasks. What does the TV distance metric represent in this setting, and what exactly does the probability mass graph suggest?
>
> A. Regarding the TV distance metric, we have revised the axis label:
>
> **Before:** TV distance
> **After:** TV distance to uniform over valid colorings
>
> For the interpretation of the probability mass plot, we add an explicit explanation:
>
> **Added:** The probability mass plot illustrates how the empirical distribution over valid colorings compares to the target uniform distribution. We observe that CoT produces a distribution that is closer to uniform, whereas the looped model concentrates probability mass on a smaller subset of solutions. This indicates that CoT achieves more uniform coverage of the valid colorings.
>
> > Q1. It seems that the stochastic setting dominates in practice, does the result in this paper suggest that CoT can perform better than latent CoT under this setting?
>
> A. Our results suggest that this is indeed the case, particularly for problems that cannot be solved reliably within a single inference. In practice, stochastic methods such as majority voting have been shown to be effective. This can be viewed as a form of randomized computation. Our analysis suggests that CoT can naturally exploit this mechanism.

---

> > ### Author Rebuttal · Reviewer_y1Ks · 2026-04-03
> >
> > Thank the authors for the reply. My concerns have been fully resolved. Therefore, I decide to maintain my score.

---

### Official Review · Reviewer_pbjH · 2026-03-10

**Soundness:** 3
**Presentation:** 3
**Significance:** 2
**Originality:** 2
**Overall Recommendation:** 4
**Confidence:** 3

**Summary:**

This paper provides a theoretical comparison between chain-of-thought (CoT) reasoning and latent thought reasoning (e.g., looped Transformers or Coconut). Overall, a central concept explored by this study is the computational tradeoff between explicit token-level reasoning and latent hidden-state reasoning when Transformers are used iteratively. The work claims to analyze a relevant issue: understanding the theoretical capabilities and limitations of different reasoning paradigms used in modern language models.

Using circuit complexity analysis, the authors show that latent thought models can efficiently simulate parallel computation, achieving expressive power corresponding to classes such as AC^{k} and TC^{k} with polylogarithmic iterations, whereas CoT reasoning is inherently sequential and therefore weaker in this regime. Conversely, the paper argues that CoT models can leverage stochastic token generation to emulate randomized algorithms, enabling approximate counting and sampling procedures that deterministic latent reasoning cannot achieve under the same assumptions. Experiments on synthetic algorithmic tasks are provided to illustrate these theoretical claims.

**Compliance With Llm Reviewing Policy:**

Affirmed.

**Key Questions For Authors:**

- The separation in approximate counting relies on the stochastic nature of token generation in CoT. Could latent models also inject randomness, e.g. incorporate stochastic hidden-state transitions or noise injection? If so, would the theoretical separation still hold?


- Recent work proposes hybrid reasoning methods combining latent and explicit reasoning. Do the authors expect such models to inherit the advantages of both paradigms, or would the theoretical tradeoffs identified here still apply?

**Limitations:**

yes

**Strengths And Weaknesses:**

**Strengths:**

*Addresses an important conceptual question*

Understanding the tradeoff between explicit reasoning traces and latent reasoning is an important problem for future LLM architectures and inference-time reasoning methods.

*Clear and well-structured theoretical analysis*

The paper presents a clean theoretical framework for comparing reasoning paradigms and connects them to classical results in computational complexity. The exposition is generally clear, and the formalization of reasoning paradigms as iterative Transformer models is well articulated.

*Balanced perspective on reasoning paradigms*

Rather than arguing that one paradigm dominates the other, the paper identifies complementary strengths: latent thought enables efficient parallel computation, while CoT supports stochastic algorithms such as approximate counting.

*Empirical illustrations supporting theoretical intuition*

The experiments on algorithmic tasks (e.g., graph connectivity, arithmetic evaluation, approximate counting) help illustrate the theoretical claims and provide some empirical grounding.

**Weaknesses**

*Incremental novelty relative to prior theoretical work*

 Much of the technical framework builds on prior work analyzing the expressivity of Transformers and CoT reasoning. The main contributions appear to extend existing circuit-complexity analyses rather than introducing fundamentally new analytical techniques. The main intuition on CoT and latent reasoning leverages prior work on parallel computation [1] and CoT expressivity analysis in [2] and subsequent works. Nevertheless, as mentioned in the strengths, this work helps formalize several aspects of the comparison and extends empirical illustrations, although many of the high-level intuitions have appeared previously.
[1] “Reasoning by Superposition: A Theoretical Perspective on Chain of Continuous Thought”
 [2] “The Expressive Power of Transformers with Chain of Thought”

*The CoT advantage may primarily arise from stochastic decoding*

Some of the separations appear to depend more on the presence or absence of randomized computation rather than intrinsic differences between reasoning paradigms.
The separation in approximate counting relies on CoT’s ability to sample intermediate tokens. However, this advantage may not be intrinsic to CoT reasoning itself; rather, it stems from allowing randomized computation. Latent reasoning models could potentially incorporate stochasticity as well, which would weaken the claimed separation. This raises the question of whether the separation reflects an intrinsic difference between CoT and latent reasoning, or primarily the distinction between deterministic and randomized computation.

*Connection to hybrid reasoning models*

The Coconut work itself uses both text and latent tokens, whereas the study in this paper focuses on the latent-only part. Recent work proposes hybrid reasoning methods combining latent and explicit reasoning or methods that perform latent looping followed by explicit CoT generation. These architectures blur the distinction between the two paradigms studied in the paper. It would be useful to clarify whether such hybrid approaches would inherit advantages from both paradigms or whether the theoretical tradeoffs identified here would still apply.

---

> ### Author Rebuttal · Authors · 2026-03-28
>
> > W1. Much of the technical framework builds on prior work analyzing the expressivity of Transformers and CoT reasoning. The main contributions appear to extend existing circuit-complexity analyses rather than introducing fundamentally new analytical techniques. The main intuition on CoT and latent reasoning leverages prior work on parallel computation [1] and CoT expressivity analysis in [2] and subsequent works. Nevertheless, as mentioned in the strengths, this work helps formalize several aspects of the comparison and extends empirical illustrations, although many of the high-level intuitions have appeared previously.
>
> A1. We would like to clarify that **our work establishes a new separation result in $TC^d$, which has not been shown in prior work** [1,2]. While our framework builds on existing perspectives, this theoretical separation constitutes a novel contribution.
>
> In addition, we would like to note that related work appeared around the same time as our initial arXiv submission (May 2025), and can be regarded as concurrent work. To avoid any confusion, we have explicitly mentioned this concurrent work in the related work section.
>
> **Added:** Concurrent work (Zhu et al., 2025a; Merrill & Sabharwal, 2025b) has also identified connections between parallel reasoning and Coconut and diffusion models.
>
> We appreciate the reviewer’s careful reading and thoughtful comments.
>
> > W2. Some of the separations appear to depend more on the presence or absence of randomized computation rather than intrinsic differences between reasoning paradigms. The separation in approximate counting relies on CoT’s ability to sample intermediate tokens. However, this advantage may not be intrinsic to CoT reasoning itself; rather, it stems from allowing randomized computation. Latent reasoning models could potentially incorporate stochasticity as well, which would weaken the claimed separation. This raises the question of whether the separation reflects an intrinsic difference.
> > Q1. The separation in approximate counting relies on the stochastic nature of token generation in CoT. Could latent models also inject randomness? If so, would the theoretical separation still hold?
>
> A2. **Even if stochastic decoding is introduced at the output token level of latent reasoning models, as shown in Figure 4, latent reasoning models still cannot realize randomized computation in the same way as CoT**. That is, current latent reasoning paradigms cannot effectively exploit randomized computation introduced by stochastic decoding, which constitutes an intrinsic difference between the two paradigms. To clarify this point, we have added the explanation in Section 4:
>
> **Before:** For latent thought, reasoning iterations are performed entirely in hidden space; no linguistic tokens are sampled except for the output token yi, as illustrated in Fig. 4.
> **After:** **We also allow stochastic decoding for latent reasoning at the token level**: reasoning iterations are performed entirely in hidden space and no linguistic tokens are sampled except for the output token $y_i$, as illustrated in Fig. 4.
>
> We thank the reviewer for this insightful comment.
>
> > W3. *Connection to hybrid reasoning models:* The Coconut work itself uses both text and latent tokens, whereas the study in this paper focuses on the latent-only part. Recent work proposes hybrid reasoning methods. These architectures blur the distinction between the two paradigms studied in the paper. It would be useful to clarify whether such hybrid approaches would inherit advantages from both paradigms.
> > Q2. Recent work proposes hybrid reasoning methods combining latent and explicit reasoning. Do the authors expect such models to inherit the advantages of both paradigms, or would the theoretical tradeoffs identified here still apply?
>
> A. We agree that whether hybrid reasoning models can inherit both advantages is an important question. We would like to clarify that raising this question is one of the key insights of our work. In particular, **this question only emerges from our characterization of the respective capabilities of CoT and latent reasoning**. Importantly, the existence of hybrid architectures does not weaken our results; rather, it highlights a key implication of our analysis.
>
> At the same time, we believe that a thorough investigation of hybrid approaches is beyond the scope of the current work and is better left for future research. Although we can offer some preliminary insights based on our current analysis. **We view diffusion language models as a promising candidate in this direction**, as noted in the conclusion section. We also note that prior work, such as Coconut, explores hybrid approaches combining CoT and latent reasoning, although we found that it presents practical difficulties in training.
>
> We thank the reviewer for highlighting the important perspective and hope this clarifies the reviewer’s question.

---

> > ### Author Rebuttal · Reviewer_pbjH · 2026-04-03
> >
> > Thanks for your response.
> > I will maintain my score.

---

### Official Review · Reviewer_7cGZ · 2026-03-12

**Soundness:** 2
**Presentation:** 2
**Significance:** 2
**Originality:** 2
**Overall Recommendation:** 5
**Confidence:** 3

**Summary:**

This paper is on the comparison of two different reasoning paradigm, i.e., chain-of-thought reasoning and latent reasoning. To be concrete, it conducts complexity-theoretic analysis. The main results are: (1) Latent thought, which enable parallel reasoning in latent space, is more preferred in the complexity class; (2) the chain-of-thought reasoning is more preferred in the sense of approximate counting.

**Compliance With Llm Reviewing Policy:**

Affirmed.

**Final Justification:**

Concerns are sufficiently addressed in rebuttal phase.

**Key Questions For Authors:**

Please see the weakness part.

**Limitations:**

Yes

**Strengths And Weaknesses:**

**Strength**

- The paper provides rigorous formalization and characterization on the two types of reasoning approach; and provide valuable insights to separate them in different problem settings.

- The empirical results also demonstrate the advantage of latent thought in parallelizable tasks. And the Chain-of-Thought has advantage in approximate counting and sampling tasks.



**Weakness and Questions**

- How would the discussed problem settings be related to more complex settings in real-world tasks? To the best of my knowledge, the paper is on comparing their capability on different types of tasks. So here is my decomposed questions:
    - In realistic reasoning tasks (like coding, math reasoning, and reasoning with scientific knowledge), how could one make decision to choose the appropriate reasoning paradigms?
    - Also, could current experiments reveals some suggestions on the future reasoning paradigm that combines the both good of the chain and latent thought?
- It would be better to include more preliminary introduction on the technical background. For example, in the section 2, the paper gives formalization on different reasoning paradigms. However, *the related concepts about Boolean circuits, and the notation about $\text{TC}^k$* are not introduced. The general readers in machine learning may not be familiar with these background, and consequently would hinder the paper's influence.

---

> ### Author Rebuttal · Authors · 2026-03-28
>
> > W1-1. How would the discussed problem settings be related to more complex settings in real-world tasks? To the best of my knowledge, the paper is on comparing their capability on different types of tasks. So here is my decomposed questions: In realistic reasoning tasks (like coding, math reasoning, and reasoning with scientific knowledge), how could one make decision to choose the appropriate reasoning paradigms?
>
> A. We thank the reviewer for this insightful question.  Our problem settings can be related to real-world problems as follows:
>
> - $TC^d$: **“Easy” problems**, which admit efficient and structured solutions (e.g., basic arithmetic such as multiplication), and
> - Counting: **“Hard” problems**, where obtaining the correct answer in a single deterministic pass is difficult (e.g., mathematical olympiad problems).
>
> Our results suggest that latent reasoning is well-suited for the former, where computation can be parallelized or efficiently structured, whereas CoT reasoning is more effective for the latter, where randomized computation, enabling exploration over multiple reasoning paths, is beneficial. To better clarify how our findings inform the choice of appropriate reasoning paradigms, we have revised the conclusion as follows:
>
> **Before:** These insights offer practical guidance for reasoning model design.
> **After:** Our results provide practical guidance for selecting between reasoning paradigms: latent reasoning is more suitable for problems that can be solved efficiently, whereas CoT is more effective for more complex problems.
>
> For tasks such as coding or knowledge retrieval, which often involve components beyond reasoning (e.g., retrieval or generation), our current framework does not directly capture all aspects of these settings. However, we can still provide some insights. For instance, problems such as recognizing context-free grammars admit parallel solutions [1], and our results suggest that latent reasoning may be advantageous in such settings.
>
> [1] Gibbons, Efficient Parallel Algorithms, 1945.
>
> > W1-2 Also, could current experiments reveals some suggestions on the future reasoning paradigm that combines the both good of the chain and latent thought?
>
> A. **We view diffusion language models as a promising candidate in this direction**, as noted in the conclusion section. We also note that prior work, such as Coconut, explores hybrid approaches combining CoT and latent reasoning, although we found that it presents practical difficulties in training.
>
> We would also like to clarify that raising this question is one of the key insights of our work. In particular, this question emerges from our characterization of the respective capabilities of CoT and latent reasoning. We thank the reviewer for highlighting this important perspective.
>
> > W2. It would be better to include more preliminary introduction on the technical background. For example, in the section 2, the paper gives formalization on different reasoning paradigms. However, the related concepts about Boolean circuits, and the notation about are not introduced. The general readers in machine learning may not be familiar with these background, and consequently would hinder the paper's influence.
>
> A. We agree that additional background on the technical concepts would improve the accessibility of the paper. To address this, **we have added a preliminary section in the appendix that introduces the necessary background**. Specifically, the new section (i) introduces Boolean circuits from basic definitions, (ii) explains the definitions of the circuit classes, including $TC^d$. We thank the reviewer for the helpful suggestion to make the paper more accessible to a broader machine learning audience.

---

> > ### Author Rebuttal · Reviewer_7cGZ · 2026-04-03
> >
> > thanks for the response.

---

### Decision · Program_Chairs · 2026-04-30

**Decision:**

Accept (regular)

**Comment:**

All reviewers agree that this is an interesting and important theoretical contribution to understanding LLM reasoning paradigms, and I join the reviewers in recommending the acceptance of this work to ICML.